## PROCEEDINGS A

### Subject Areas:
mechanical engineering, applied mathematics, chaos theory

### Keywords:
model predictive control, finite-time Lyapunov exponents, path planning, mobile sensors, dynamical systems, unsteady fluid dynamics

### Author for correspondence:
Kartik Krishna
e-mail: karkris3@uw.edu

# Finite-horizon, energy-efficient trajectories in unsteady flows

Kartik Krishna[1], Zhuoyuan Song[2] and
Steven L. Brunton[1]

[1]Department of Mechanical Engineering, University of Washington, Seattle, WA 98195, USA
[2]Department of Mechanical Engineering, University of Hawai'i at Mānoa, Honolulu, HI 98116, USA

KK, 0000-0001-5261-9882; ZS, 0000-0002-7442-8435; SLB, 0000-0002-6565-5118

Intelligent mobile sensors, such as uninhabited aerial or underwater vehicles, are becoming prevalent in environmental sensing and monitoring applications. These active sensing platforms operate in unsteady fluid flows, including windy urban environments, hurricanes and ocean currents. Often constrained in their actuation capabilities, the dynamics of these mobile sensors depend strongly on the background flow, making their deployment and control particularly challenging. Therefore, efficient trajectory planning with partial knowledge about the background flow is essential for teams of mobile sensors to adaptively sense and monitor their environments. In this work, we investigate the use of finite-horizon model predictive control (MPC) for the energy-efficient trajectory planning of an active mobile sensor in an unsteady fluid flow field. We uncover connections between trajectories optimized over a finite-time horizon and finite-time Lyapunov exponents of the background flow, confirming that energy-efficient trajectories exploit invariant coherent structures in the flow. We demonstrate our findings on the unsteady double gyre vector field, which is a canonical model for chaotic mixing in the ocean. We present an exhaustive search through critical MPC parameters including the prediction horizon, maximum sensor actuation, and relative penalty on the accumulated state error and actuation effort. We find that even relatively short prediction horizons

can often yield energy-efficient trajectories. We also explore these connections on a three-dimensional flow and ocean flow data from the Gulf of Mexico. These results are promising for the adaptive planning of energy-efficient trajectories for swarms of mobile sensors in distributed sensing and monitoring.

## 1. Introduction

The ability to generate energy-efficient trajectories that take advantage of the inherent motions of a background flow field has significant implications for monitoring large bodies of water with intelligent mobile sensors [1–3], furthering our understanding of the climate and natural ecosystems [4–6]. Developments in this area also present economic opportunities for cost reduction in industries that rely heavily on maritime transport and shipping. Self-powered mobile sensors typically have complex performance tradeoffs, limiting size, weight and power (SWAP). Furthermore, most mobile sensors will only have partial and imperfect information about the ambient flow field, resulting in a finite-horizon predictive window to make decisions about its trajectory. Improving the generation of energy-efficient trajectories that intelligently leverage the flow field to *go with the flow* may have significant benefits in extending the duration and reach of these mobile sensing platforms. This work provides an extensive analysis of trajectories generated through a finite-horizon model predictive control (MPC) optimization of a mobile sensor in a time-varying background flow across a wide range of system parameters. Furthermore, we establish connections between the control performance and efficiency with the alignment of these trajectories along coherent structures in the background flow.

Currently, there exists an extensive literature that has investigated various algorithms for trajectory generation for such transport applications. For example, graph search algorithms and stochastic optimization have been investigated for path planning [7–9]. Assimilating *in situ* observations obtained by mobile sensors in an adaptive fashion into ocean models has also been explored, for example with mixed integer programming algorithms [10,11]. Coordinated control of ocean gliders for adaptive ocean sensing has been exhaustively studied in Monterey Bay [12–15]. Algorithms inspired from computational fluid dynamics have also been used to explore coordinated control of swarms in flow fields [16–20]. More recent developments include the use of reinforcement learning algorithms to find optimal paths in turbulent flow fields [21–23]. However, there has been relatively little work in developing a deep understanding of the connection between the dynamics of the flow field and the nature of the optimal trajectories within the flow fields, with a few notable exceptions [24–27]. A key challenge in exploring this connection is the complexity of fluid flow fields, which typically involve the existence of multiple scales in space and time.

To understand the complexity of fluids, techniques from dynamical systems are often employed. Lagrangian coherent structures (LCS) have emerged as a robust and principled approach to uncover invariant manifolds that mediate the transport of material in unsteady fluid flows [28–34]. Specifically, LCS define the transport barriers in a flow field where passive drifters are attracted to or repelled by. There has been considerable work in the development of algorithms to accurately and efficiently compute these structures from data [30,33,35–42]. The finite-time Lyapunov exponent (FTLE) is a scalar value that characterizes the divergence from a trajectory over a finite time interval, and is often used to compute LCS. The FTLE method has been successfully applied to domains of bio-propulsion [43], medicine [44,45], the spread of microbes [46] and the study of aerodynamics [47,48].

The ideas from both trajectory generation and the theory of LCS have been related in the past [24–27]. A predecessor of this was the planning of space missions using invariant manifolds [49]. In the context of ocean transport, Inanc *et al.* [24] showed that the optimal trajectories of autonomous agents generated using a receding-horizon optimal control algorithm overlap with LCS. Moreover, Senatore & Ross [26] exploited this idea further to generate energy optimal paths

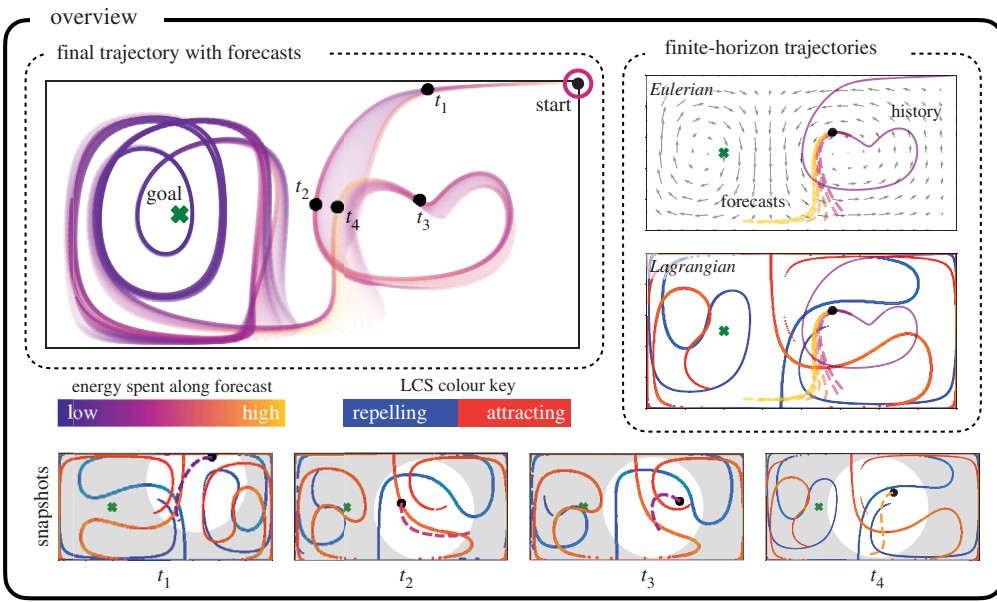

**Figure 1.** Overview of the proposed methodology for analysing the connections between finite-horizon energy optimal trajectories and the FTLE field. A self-propelling agent is controlled to transit from a starting location to a goal location through a finite-time horizon energy-optimal trajectory in a time-varying double gyre flow field. The resulting agent trajectory, along with the finite-horizon predicted trajectories at each time step, are shown and colour-coded based on instantaneous energy expenditure (top left). The trajectory history (solid) and the future forecast trajectory bundle (dashed) at an example time instant are shown (top right); the instantaneous FTLE ridges are also shown below these with blue indicating the repelling LCS and red indicating the attracting LCS. As can be observed from the snapshots taken at four particular times (bottom), the energy expenditure along the planned trajectory, given by the colour of the dashed line, and the shape of the finite-horizon trajectory depend on the evolution of the local FTLE ridges. (Online version in colour.)

by controlling the agents to track the background LCS. Recent papers have further explored the connections between optimal control and LCS [50–52] in the context of path planning in the ocean. However, there is still a need to better understand how the prediction horizon and relative cost of actuation in the autonomous agent optimization relate to the use of coherent structures in the unsteady background flow.

In this work, we investigate the explicit connections between finite-horizon energy-optimal trajectories of a mobile sensor and the underlying background flow dynamics. We specifically analyse how key parameters of the MPC-based optimization affect how the resulting autonomous agent trajectory uses unsteady fluid coherent structures for energy-efficient transport. This analysis is performed primarily on the double gyre flow field, which is a testbed to understand mixing and transport in the ocean. Subsequently, we also further verify our observations on advanced flow fields including the Arnold–Beltrami–Childress (ABC) flow, a three-dimensional incompressible analytical flow field, and unsteady flow data from the Gulf of Mexico. A summary of our methodology is shown in figure 1. The choice of MPC is particularly relevant in this work, as both the FTLE and MPC rely on finite-time horizons in their computations. To explore this connection, we perform an exhaustive search through several of the trajectory optimization parameters that are important to practitioners, including the prediction time horizon and step size for MPC, the relative cost of actuation versus state tracking error, and the maximum agent velocity. We find that there are strong correlations between the presence of background FTLE ridges and the actuation energy expenditure at the corresponding locations along the trajectory.

The remainder of this work is organized as follows. In §2, the core methodologies of MPC and FTLE are discussed. MPC will be the primary optimization algorithm used to generate trajectories, and these will be analysed using FTLE fields. Section 3 describes models for the mobile sensor dynamics and actuation, along with the dynamics of the unsteady double gyre background flow field. The main results are presented in §4, including in-depth analysis of trajectories generated across a wide range of system parameters. In particular, the time horizon of the MPC optimization, the relative cost of actuation versus state tracking error, and the frequency of the background flow oscillation are all investigated. In §5, we highlight the use of MPC on other flow fields and demonstrate our observations on the ABC and Gulf of Mexico flow fields. Section 6 provides a summary of results and a discussion of limitations with suggestions for future work. Appendix A also provides additional plots and analysis of the data that was not presented in the main text.

## 2. Methodology

In this section, we introduce two approaches for analysing and generating fuel-efficient trajectories for a mobile sensor in an unsteady background flow: FTLE fields and MPC. First, we introduce the computation of FTLE fields [28,30,33] for passive tracer particles to extract LCS from a time-varying flow field. This method is particularly important to characterize the uncontrolled behaviour of drifters in terms of finite-time attraction and repulsion behaviours. Next, we introduce the preliminaries of finite-horizon MPC, which is an online control optimization algorithm that optimizes a cost function defined over a finite-time prediction horizon. We will use MPC for trajectory optimization of a mobile sensor in an unsteady background flow. MPC is a natural choice, since the mobile sensor will have limited actuation authority, and information about the flow field will only be approximate as it is limited to a finite-time horizon.

## (a) Finite-time Lyapunov exponents

Given a vector field $\mathbf{v}(\mathbf{x}(t), t) : \mathbb{R}^n \times \mathbb{R} \to \mathbb{R}^n$, the dynamics of a passive drifter is given by

$$\frac{\mathrm{d}}{\mathrm{d}t}\mathbf{x}(t) = \mathbf{v}(\mathbf{x}(t), t). \tag{2.1}$$

Here, $t \in \mathbb{R}$ represents time, and $\mathbf{x}(t) \in \mathbb{R}^n$ is the position of the drifter, where typically $n = 2$ or 3, depending on the dimension of space. The FTLE field can be used to determine the LCS of an unsteady vector field [28,30,33]. The LCS are curves or surfaces in the domain where nearby trajectories $\mathbf{x}(t)$ are strongly attracted to or repelled from, making them time-varying analogues of stable and unstable invariant manifolds in dynamical systems theory [53].

The FTLE algorithm is as follows. First, a grid of drifters is initialized at time $t_0$ and numerically integrated through the flow field $\mathbf{v}(\mathbf{x}(t), t)$ for a fixed amount of time (i.e. the time horizon) $T \in \mathbb{R}$, resulting in a flow map $\boldsymbol{\Phi}_{t_0}^{t_0+T} : \mathbb{R}^n \to \mathbb{R}^n$:

$$\boldsymbol{\Phi}_{t_0}^{t_0+T} : \mathbf{x}(t_0) \mapsto \mathbf{x}(t_0) + \int_{t_0}^{t_0+T} \mathbf{v}(\mathbf{x}(\tau), \tau) \, \mathrm{d}\tau. \tag{2.2}$$

The flow map operator $\boldsymbol{\Phi}_{t_0}^{t_0+T}$ takes each drifter at an initial condition $\mathbf{x}(t_0)$ and returns its new position $\mathbf{x}(t_0 + T)$ after it is advected through the vector field for a time $T$. Next, the Jacobian matrix of partial derivatives of the flow map, $\mathbf{D}\boldsymbol{\Phi}_{t_0}^{t_0+T}$, is computed using finite differences for

each drifter in the grid, represented by the grid node indices $i, j \in \mathbb{Z}^+$, such that

$$(\mathbf{D}\boldsymbol{\Phi}_{t_0}^{t_0+T})_{i,j} = \begin{bmatrix} \dfrac{\Delta x_i(t_0+T)}{\Delta x_i(t_0)} & \dfrac{\Delta x_j(t_0+T)}{\Delta y_j(t_0)} \\[2ex] \dfrac{\Delta y_i(t_0+T)}{\Delta x_i(t_0)} & \dfrac{\Delta y_j(t_0+T)}{\Delta y_j(t_0)} \end{bmatrix}$$

$$= \begin{bmatrix} \dfrac{x_{i+1,j}(t_0+T) - x_{i-1,j}(t_0+T)}{x_{i+1,j}(t_0) - x_{i-1,j}(t_0)} & \dfrac{x_{i,j+1}(t_0+T) - x_{i,j-1}(t_0+T)}{y_{i,j+1}(t_0) - y_{i,j-1}(t_0)} \\[2ex] \dfrac{y_{i+1,j}(t_0+T) - y_{i-1,j}(t_0+T)}{x_{i+1,j}(t_0) - x_{i-1,j}(t_0)} & \dfrac{y_{i,j+1}(t_0+T) - y_{i,j-1}(t_0+T)}{y_{i,j+1}(t_0) - y_{i,j-1}(t_0)} \end{bmatrix}, \quad (2.3)$$

where $x, y \in \mathbb{R}$ are the horizontal and vertical components of the position vector $\mathbf{x}(t)$. This flow map Jacobian is used to compute the Cauchy–Green deformation tensor, given by

$$\boldsymbol{\Delta}_{i,j} = (\mathbf{D}\boldsymbol{\Phi}_{t_0}^{t_0+T})^* \mathbf{D}\boldsymbol{\Phi}_{t_0}^{t_0+T}, \quad (2.4)$$

where $*$ represents the matrix transpose, not to be confused with the duration of integration $T$. Finally, the largest eigenvalue $\lambda_{\max}$ of $\boldsymbol{\Delta}_{i,j}$ for each drifter $i, j$ is used to compute the FTLE field:

$$\sigma_{i,j} = \frac{1}{|T|} \ln \sqrt{(\lambda_{\max})_{i,j}}. \quad (2.5)$$

Alternatively, $\sigma_{i,j}$ can be computed as the largest singular value from the singular value decomposition of $\mathbf{D}\boldsymbol{\Phi}_{t_0}^{t_0+T}$. It is important to note that for unsteady flow fields, the FTLE field will also vary in time, so that at each new time step a new grid of particles must be reinitialized and advected through the flow. This procedure is typically quite expensive, although there are algorithms to eliminate redundant calculations [35,36].

LCS are often computed as ridges of the FTLE field, which requires an additional step of computing the Hessian of $\sigma_{i,j}$ for ridge extraction. FTLE based on drifters integrated forward in time, $T > 0$, results in coherent structures that repel drifters. Similarly, FTLE based on drifters integrated backward in time, $T < 0$, results in coherent structures that attract drifters. These can be seen in figure 1 as red and blue curves, where the red curves are attracting and the blue are repelling. FTLE fields and the resulting LCS are related to almost invariant sets from statistical dynamical systems [54–57]. In particular, LCS act as separatrices in the flow, segmenting different regions where passive tracers remain trapped [58]. FTLE and LCS have also been used extensively to analyse ocean flows [59–61], for example to model the spread of pollution [62]. More broadly, FTLE has also been used to compute coherent structures for a wide range of other flows [63–68]. In this work, we will use FTLE fields generated from passive particles to investigate the trajectories of active mobile sensors, to understand how and when these sensors exploit structures in the flow field for energy-efficient transport.

## (b) Model predictive control

The dynamics of mobile sensors operating in real environments are often strongly nonlinear and subject to hardware constraints, time delays, non-minimum phase dynamics, instability, and restrictions on actuation capability. These limitations make the use of traditional linear control approaches challenging, motivating the powerful MPC optimization [69–72] described here. In this work, we use MPC to generate trajectories for a mobile sensor in an unsteady background flow and investigate how these trajectories vary with the optimization parameters.

In general, the dynamics of a nonlinear system with actuation $\mathbf{u} \in \mathbb{R}^m$ can be written as

$$\frac{\mathrm{d}}{\mathrm{d}t}\mathbf{x}(t) = \mathbf{g}(\mathbf{x}(t), \mathbf{u}(t), t), \quad (2.6)$$

where $\mathbf{g}$ is the controlled vector field. In the context of this paper, the state $\mathbf{x}$ can be either the position of the agent, as in the previous section, or both the position and velocity of the agent.

MPC is a powerful method for calculating the actuation $\mathbf{u}$ by formulating an iterative optimization problem that minimizes a cost function over a finite-time horizon. The controller enacts this optimized actuation policy for a short time, often for a single time step, and then the optimization problem is recomputed and initialized at the current state. In this way, MPC is quite robust to model uncertainty and disturbances, as the optimization is continuously being reinitialized as new information is available about how the system actually responds to the actuation. Computing over a finite-time horizon might also make MPC more flexible and faster than a global optimization technique, especially for chaotic systems, which may result in stiff long-time optimizations. These benefits make MPC more versatile and widely used over other traditional trajectory generation algorithms. Finally, the FTLE and MPC computations are both performed over a finite time horizon, suggesting the potential for a connection between the outputs of the two algorithms.

Typically, the optimization cost for MPC can be formulated as

$$J = \mathbf{e}(t_0 + T_H)^T \mathbf{Q}_2 \mathbf{e}(t_0 + T_H) + \int_{t_0}^{t_0 + T_H} [\mathbf{e}(\tau)^T \mathbf{Q}_1 \mathbf{e}(\tau) + \mathbf{u}(\tau)^T \mathbf{R} \mathbf{u}(\tau)] d\tau, \tag{2.7}$$

subject to the system dynamics in (2.6) and control constraints imposed by physical limitations:

$$\mathbf{u}_{\min} \leq \mathbf{u}(t) \leq \mathbf{u}_{\max}. \tag{2.8}$$

Here, $\mathbf{u}_{\min}$ and $\mathbf{u}_{\max}$ are the minimum and maximum values the components of $\mathbf{u}$ can take, respectively. For example, the actuators may be unable to produce thrusts beyond a certain value. The state error is given by $\mathbf{e}(t) = \mathbf{x}(t) - \mathbf{x}_{\text{goal}}$. The finite-time horizon over which we forecast our model for the optimization is $T_H \in \mathbb{R}^+$; this term is similar to $T$, the advection time used to calculate FTLE. $\mathbf{R} \in \mathbb{R}^{m \times m}$ is a positive definite matrix that quantifies the penalty on actuation effort, and $\mathbf{Q}_1 \in \mathbb{R}^{n \times n}$ and $\mathbf{Q}_2 \in \mathbb{R}^{n \times n}$ are positive semi-definite matrices that quantify the penalty on deviations of the state from the goal throughout the trajectory and at the final time step, respectively. For computational purposes, (2.7) is often discretized. The sampling time step is $\Delta t$, the discretization of $d\tau$. In this paper, the $\Delta t$ serves as the 'replanning frequency', or, the time after which the a re-optimization of a trajectory is carried out. It is possible to improve the computational speed and convergence of the algorithm with a *warm start*, which uses the trajectory computed in a previous instance as the initial guess for the trajectory in the next instance [73].

## 3. Model problem

We now discuss the models used to simulate the agent dynamics and the unsteady flow field the mobile sensor operates within. We also provide specific parameters that are used for all numerical experiments.

### (a) Sensor dynamics

In a two-dimensional setting, a simple kinematic model for the dynamics of the mobile sensor is given by adding the velocity due to actuation, $\mathbf{u}(t)$, to the background flow velocity $\mathbf{v}(\mathbf{x}(t), t)$: $\mathbb{R}^2 \times \mathbb{R} \to \mathbb{R}^2$:

$$\frac{d}{dt} \mathbf{x}(t) = \mathbf{v}(\mathbf{x}(t), t) + \mathbf{u}(t). \tag{3.1}$$

The state $\mathbf{x}(t) = [x, y] \in \mathbb{R}^2$ is the position vector. The key assumption in this model is that, without control, the velocity of the sensor, $d\mathbf{x}/dt$, matches the velocity of the background fluid flow. Thus, the uncontrolled mobile sensor can be considered as a passive Lagrangian drifter, and (3.1) degenerates to (2.1) when $\mathbf{u}(t) = 0$ all times. Moreover, it assumes that the sensor can generate its own relative velocity $\mathbf{u}(t) = [u_x, u_y] \in \mathbb{R}^2$ in addition to the flow-induced velocity. It is possible to develop more sophisticated models for the mobile sensor dynamics that include inertial and

rotational dynamics; in Zhang *et al.* [25], it was shown that trajectories based on such models also show strong correlation with the presence of background LCS.

## (b) Double gyre flow field

We will investigate the motion of the mobile sensor above in the unsteady double gyre flow field described here. The double gyre flow is an analytically defined, periodic vector field that is often used to study mixing and coherent structures related to those found in geophysical circulations. In particular, the double gyre represents a typical large-scale ocean circulation phenomenon often observed in the northern mid-latitude ocean basins. This circulation is quite dominant and is persistent, consisting of sub-polar and sub-tropical gyres. As a major type of ocean circulation, several main features of the double gyre phenomena have been identified through analysing observational data and numerical simulations [74–76].

The double gyre velocity field is derived from the stream function

$$\phi(x, y, t) = A \sin(\pi f(x, t)) \sin(\pi y), \tag{3.2}$$

where the time dependency is introduced by

$$f(x, t) = a(t)x^2 + b(t)x, \tag{3.3}$$

with time-dependent coefficients

$$a(t) = \epsilon \sin(\omega t) \quad \text{and} \quad b(t) = 1 - 2\epsilon \sin(\omega t).$$

This flow is defined on a non-dimensionalized domain of $[0, 2] \times [0, 1]$, where $A$, $\epsilon$, $\omega$, $x$, $y$, $t \in \mathbb{R}$. Here, $\epsilon$ dictates the magnitude of oscillation in the $x$-direction, $\omega$ is the angular oscillation frequency, and $A$ controls the velocity magnitude. Unless stated otherwise, the parameters used for the double gyre flow field are as in Shadden *et al.* [30], where $A = 0.1$, $\epsilon = 0.25$ and $\omega = 2\pi/10$. The resulting velocity field is given by

$$\mathbf{v}(x, y, t) = \begin{bmatrix} -\dfrac{\partial \phi}{\partial y} \\ \dfrac{\partial \phi}{\partial x} \end{bmatrix} = \begin{bmatrix} -\pi A \sin(\pi f(x, t)) \cos(\pi y) \\ \pi A \cos(\pi f(x, t)) \sin(\pi y) \end{bmatrix}. \tag{3.4}$$

## (c) Specific control objective

By combining the mobile sensor model and the double gyre flow field, the dynamics of the sensor are given by

$$\frac{\mathrm{d}}{\mathrm{d}t} \begin{bmatrix} x \\ y \end{bmatrix} = \begin{bmatrix} -\pi A \sin(\pi f(x, t)) \cos(\pi y) \\ \pi A \cos(\pi f(x, t)) \sin(\pi y) \end{bmatrix} + \begin{bmatrix} u_x \\ u_y \end{bmatrix}. \tag{3.5}$$

The objective is to move a mobile sensor from a starting location at coordinates $\mathbf{x}_{\text{start}} = [2, 1]$ to a goal location at $\mathbf{x}_{\text{goal}} = [0.5, 0.5]$. The cost function is given by

$$J = \int_{t_0}^{t_0 + T_H} [\mathbf{e}(\tau)^T \mathbf{Q}\mathbf{e}(\tau) + \mathbf{u}(\tau)^T \mathbf{R}\mathbf{u}(\tau)] \, \mathrm{d}\tau, \tag{3.6}$$

where $\mathbf{e}(t) \triangleq \mathbf{x}(t) - \mathbf{x}_{\text{goal}}$ is the state tracking error. This cost function is subject to constraints on the actuation of the mobile sensor

$$|u_x| \leq 0.1 \quad \text{and} \quad |u_y| \leq 0.1,$$

which ensure that the maximum sensor velocity is significantly smaller than the largest background flow field velocity, $\pi A \approx 0.314$. This constraint is imposed to model the limited actuation available in real world scenarios. Here, $\mathbf{Q} = Q\mathbf{I}_{2\times 2}$ and $\mathbf{R} = R\mathbf{I}_{2\times 2}$, where $\mathbf{I}$ is the identity matrix. The discretized time step across all double gyre simulations is $\Delta t = 0.1$. Each simulation was run for 800 steps. In the following sections, we will vary the relative cost of actuation versus

state error, given by the ratio $R/Q$, and analyse how this impacts the mobile sensor trajectories. To solve the resulting optimization problems, we use the CasADi [77] and MPCTools [78] packages, which use an interior point filter-line search algorithm (IPOPT).

## 4. Double gyre results

In this section, we examine energy-efficient trajectories for an active mobile sensor generated using MPC across a range of hyperpameters, including the prediction horizon, penalty weights on the state error and control effort, and the double gyre oscillation frequency. The MPC results for this thorough parameter sweep are presented in §4(d) and summarized in figure 7. Our goal is to understand the sensitivity of the trajectory to parameters and to uncover performance tradeoffs, for example with the time horizon of optimization. We find a large *sweet spot* where effective, energy-efficient trajectories are generated. Furthermore, we establish connections between the efficient mobile sensor trajectories and the LCS of the underlying flow field.

### (a) Trajectories with different relative actuation cost, $R/Q$

Figure 2 shows the effect of varying the ratio of control effort penalty $R$ to the state error penalty $Q$ on the trajectories, for a fixed time horizon of $T_H = 4$; similar plots for a range of time horizons from $T_H = 1$ to $T_H = 12$ are shown in electronic supplementary material, figures S1–S9. The ratio $R/Q$ quantifies the relative cost of actuation, and varying this parameter is important to understand performance tradeoffs when the mobile sensor has a limited actuation budget. As $R/Q$ is increased, corresponding to actuation being more expensive, the agent actuates less, and the state tracking error increases. This increase in state tracking error tends to correspond to larger steady-state limit cycles about the goal state. The weighted actuation cost $J_u = \sum Ru^T u \Delta t$ increases with $R/Q$, as we fix $Q = 1$ and increase $R$; however, the unweighted actuation $\sum u^T u \Delta t$ decreases with $R/Q$. Importantly, the trend of cost versus $R/Q$ is not strictly monotonic, and there are discontinuous jumps corresponding to bifurcations in the orbit; the non-monotonic behaviour and bifurcations are more pronounced for other $T_H$ in appendix A. For small $R/Q$ values such as $R/Q = 2$ and $R/Q = 3$, the agent moves around the goal state in a tight orbit, and this orbit continuously expands as $R/Q$ increases, as shown for $R/Q = 15$. However, between $R/Q = 25$ and $R/Q = 26$ the trajectory undergoes a rapid qualitative change, where the radius of the orbit around the goal state jumps.

It is interesting to note in figure 2 that the $R/Q = 2$ agent has an initial loop in the right basin, while the $R/Q = 3$ agent does not. This behaviour is counterintuitive, as the $R/Q = 2$ agent should expend control more freely, and thus more aggressively seek the goal state. As shown in figure 15 in appendix A, the more aggressive agent does move away from the starting state faster initially; however, it becomes trapped on the side of a repelling LCS farther away from the goal location and must make an entire orbit around the right gyre before approaching the goal state. The maximum agent velocity is smaller than the maximum gyre velocity, so even the most aggressive agents are unable to break out of the right gyre without precise timing. This type of bifurcation also occurs for fixed $R/Q = 2$ by varying the time horizon, as in figure 3. In this case, the behaviour is more consistent with intuition, as the longer time horizon trajectories avoid being trapped in the right gyre.

Previous work [24] suggests that low-energy trajectories tend to coincide with the LCS of the background flow. In our example, even for $R/Q = 2$ and $R/Q = 3$, the mobile agent can be seen aligning with and exploiting the coherent structures. For example, in the top left of figure 2, the $R/Q = 2$ sensor moves along on the intersection of the attracting and repelling LCS as it orbits the goal state. In the next section, we will see that the agent also precisely times its actuation before and after crossing a repelling LCS to take advantage of the background drift.

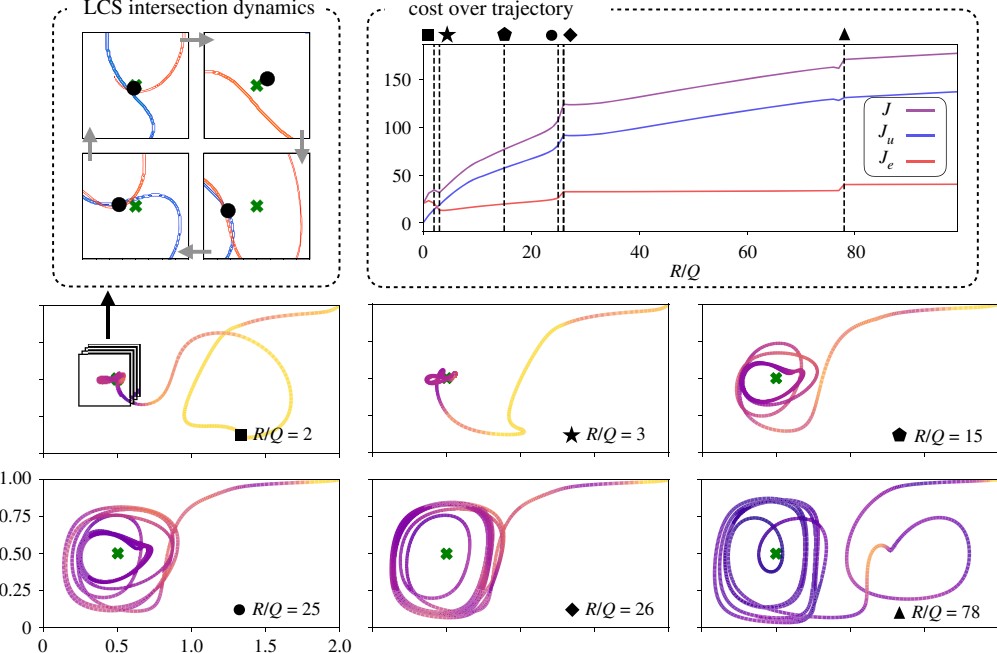

**Figure 2.** Dependency of the resulting sensor trajectories on the ratio between the penalties on energy expenditure ($R$) and state error ($Q$), for a fixed time horizon $T_H = 4$. The trajectories are colour-coded by instantaneous energy expenditure. There are three costs shown in the top right figure: $J = J_u + J_e$, $J_e = \sum Q(x - x_g)^T(x - x_g)\Delta t$, and $J_u = \sum Ru^Tu\Delta t$. Here, $J$ generally increases with $R/Q$. The trajectories undergo several qualitative changes (bifurcations) as $R/Q$ is increased, forming different types of periodic orbits shown on the bottom of the figure. An example of this is when $R/Q$ is changed from 25 to 26, we observe a major change in the shape of the final orbit around the goal, as opposed to the minor change from $R/Q = 15$ to 25, where the final eye-shaped orbit only gradually increases in size. The formation of these orbits is dependent on the background flow FTLE, as can be seen from the inset of case $R/Q = 2$. We observe that as the LCS move, the stable and unstable LCS intersect at a point, whose location changes every instant, and the sensor moves in a manner in which it is right on top of this intersection point for most of the time when $R/Q = 2$ (top left box). (Online version in colour.)

## (b) Instantaneous energy versus finite-time Lyapunov exponent ridge

Given the existing connection of low-energy trajectories and FTLE ridges, we are interested in how the energy is used along a trajectory. Figure 4 shows how the agent 'schedules' an increase in actuation to cross a repelling (blue) FTLE ridge. After crossing, the agent decreases its actuation, as it is naturally repelled from the blue ridge and attracted by the red ridge into the left basin. Similar timing and utilization of the FTLE ridges is observed for a wide range of time horizons and control aggressiveness. This occurs in the double gyre only across the *repelling* hyperbolic LCS near the middle of the domain, as opposed to the shear-driven LCS near the goal location. This can be observed across the all gyre trajectories, where the yellow (high energy) regions in the trajectory are in the middle of the domain.

## (c) Periodic orbits

We observe that controlled trajectories often form periodic orbits around the goal state, as seen in figures 2, 5 and 6. Because the background flow field is periodic, the agent would require constant actuation to stay fixed at the goal state. Instead, the agent trajectory tends to form a periodic orbit around the goal, balancing state tracking error and control expenditure. Typically, this orbit is larger for agents with a tighter energy budget (i.e. for larger $R/Q$). Many past studies

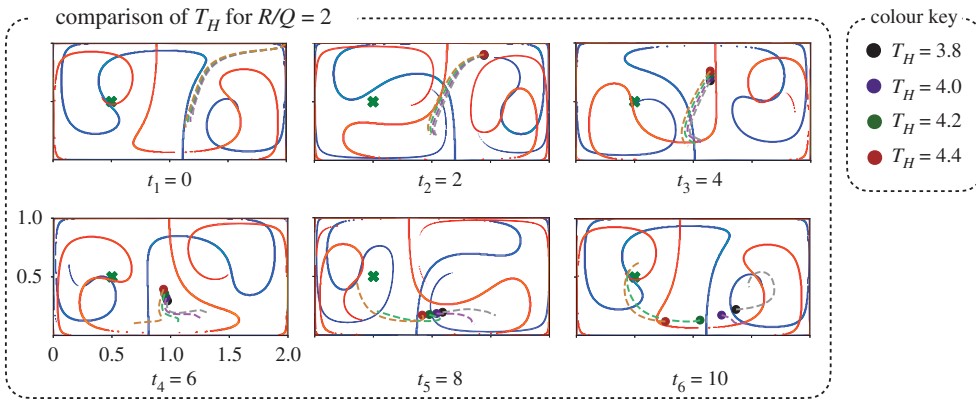

**Figure 3.** Different instances of a trajectory for different time horizons while keeping $R/Q = 2$ fixed. By varying the time horizon, $T_H$, we see that the extra loop in figure 2 is due to a sensitivity of the planned path with respect to $T_H$, where the agent becomes stuck in the right gyre for lower $T_H$. We can see here with reference to figure 15 how as the time horizon increases, the agent moves less aggressively and improves its timing with the gyre oscillations to reach the goal sooner without taking an extra loop. Another interesting aspect of this plot can be seen at $t_3 = 4$ and $t_4 = 6$ where the multiple agents line up to mirror the movement of the red LCS, further confirming the strong correlation finite-horizon optimal trajectories and the FTLE ridges. (Online version in colour.)

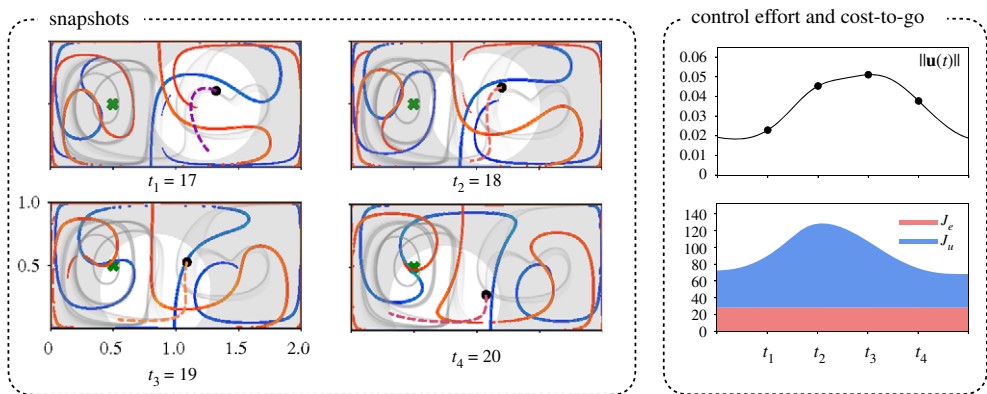

**Figure 4.** Influence of the FTLE ridges on the energy expenditure both along prediction horizon (right top) and instantaneously (right bottom) under parameters $T_H = 4.0$, $R/Q = 100$ and $\Delta t = 0.1$. An agent's motion along with the predicted forecast trajectory (dashed line) are shown on the left, together with the repelling (blue) LCS and the attracting (red) LCS. There is a correlation between the spike in both the instantaneous energy spent (right top) and the cost along the forecast trajectory (right bottom) with movement across an FTLE ridge. Here, unlike in figure 2, the summations $J_e = \sum Q(x - x_g)^T(x - x_g)\Delta t$ and $J_u = \sum Ru^T u\Delta t$ are only along the forward dashed line in the plots under 'snapshots' and not along the entire trajectory as in figure 2. (Online version in colour.)

have focused on trajectory planning where the final state is fixed at the goal. However, given the constantly evolving background flow field and its dominant effect on mobile sensor dynamics, it is important in practice, to consider the cases where the final state cannot be fixed. Figure 6 also indicates that the shape of the final periodic orbit depends on the frequency of the double gyre oscillation, with the frequency of the agent orbit synchronizing with the gyre frequency. In the other example flows observed below, similar periodic orbits are observed, where the agent *loiters* around the goal state. It will be interesting to investigate these orbits in more detail, including the classes of flows they exist in, and the conditions under which they bifurcate.

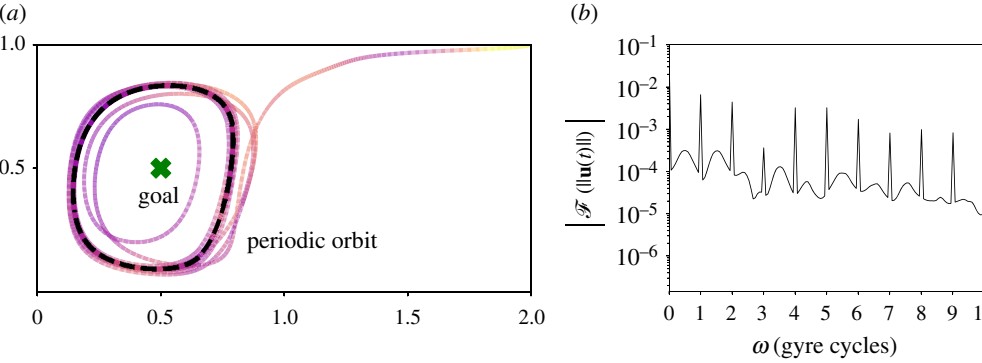

**Figure 5.** The mobile sensor settle on periodic orbits around the goal state (*a*) and the magnitude of the Fourier transform of the instantaneous energy spent by the mobile sensor (*b*). We observe that the time series of the energy spent is periodic with frequencies at integer multiples of the double gyre oscillation frequency, which correspond to the peaks in (*b*). (Online version in colour.)

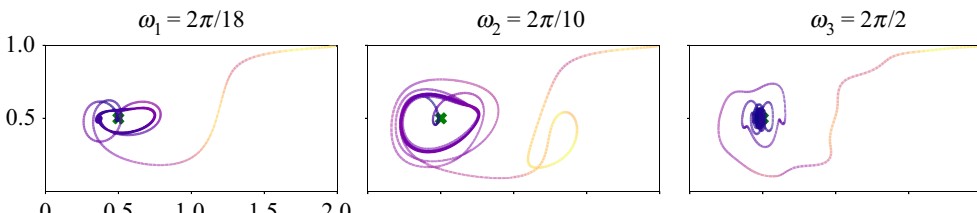

**Figure 6.** Variations of agent trajectory under different double gyre oscillation frequencies. The frequency of the periodic orbits depends on the double gyre oscillation frequency. (Online version in colour.)

## (d) Model predictive control parameter sweep

We now present an exhaustive sweep through two of the most critical parameters for MPC, the prediction horizon $T_H$ and the cost function penalty ratio $R/Q$, for different gyre oscillation frequency $\omega$. The first two parameters are related to the power and prediction capability of the mobile sensor, and the third parameter characterizes the unsteadiness of the background flow. We perform a full parameter sweep for the time horizon ($T_H \in [0, 10]$) and the cost penalty ratio $R/Q \in [0, 100]$, for the double gyre frequency $\omega \in [\pi/6, \pi/3]$. For each parameter value, we compute the state tracking error and the (unweighted) actuation energy expenditure, integrated along the entire trajectory.

Figure 7 shows the results from the MPC parameter sweep. For all time horizons and gyre frequencies, we observe that the trajectories sweep out a Pareto front in control expenditure versus state tracking error as $R/Q$ is varied logarithmically from 0 to 100. The bottom row of figure 7 shows three representative trajectories along the Pareto front. As $R/Q$ is increased, there is often a sharp drop in control cost with a relatively small increase in state tracking error, suggesting that there are energy-efficient trajectories that achieve relatively good tracking performance. However, we observe a break point in this monotonic trend, beyond which increasing $R/Q$ results in rapid deterioration of the state error with relatively little decrease in control cost. This break point corresponds to the scenario where the motion of the sensor is dominated by the background flow, and the chaotic nature of the flow field dominates the state and energy errors. This phenomenon is more evident for smaller time horizons.

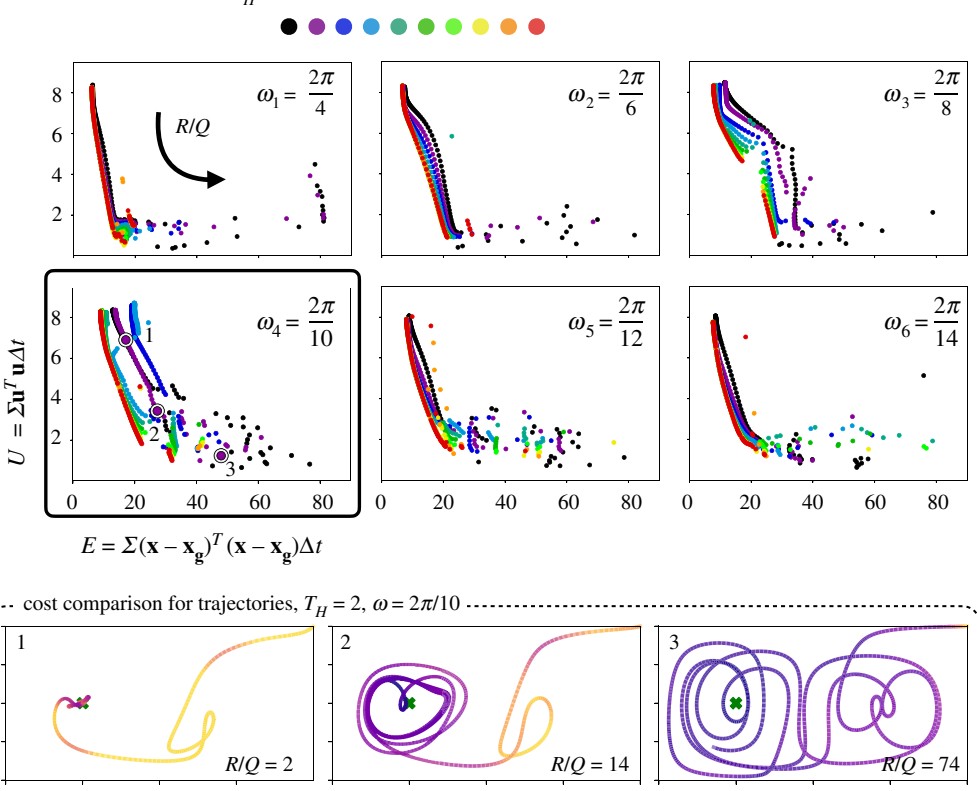

**Figure 7.** Multiple simulations were carried out at each $R/Q$ ratio spaced logarithmically, from 0 to 100, time horizon ranging from 1 to 10, and gyre frequency ranging from $2\pi/4$ to $2\pi/14$. The data presented here are in the form of scatter plots for each gyre frequency with each colour representing the Pareto optimal tradeoff curve between the total energy spent along each trajectory and the sum of deviations from target along the trajectory. The trajectories shown in the bottom row correspond to the highlighted purple circles (1,2,3) in the Pareto optimal corresponding to $\omega_4 = 2\pi/10$. (Online version in colour.)

It is observed that longer prediction horizons produce trajectories that are more energy efficient with smaller state errors. This is expected, as longer time horizons include more information about the flow field in the optimization. This trend is weaker for small-to-moderate $R/Q$ and is more pronounced for larger $R/Q$. The shape of the Pareto curve also changes with the double gyre frequency. This shape change is particularly evident for moderate frequencies, suggesting a 'resonance' in the interaction of trajectories with the background flow. Resonance with changing gyre frequency has been explored in the context of inertial particles in the double gyre flow [34].

## (e) Sensor velocity

To gain further insight into the dependency of sensor actuation velocity on the background flow velocity, we compare their distributions along the resulting trajectory at different $R/Q$ values. Figure 8 shows histograms of the magnitude and orientation of the sensor actuation velocity versus the background flow velocity, for a range of $R/Q$ values. It can be observed that more aggressive agents with smaller $R/Q$ have larger actuation velocity magnitudes and tend to move perpendicular to the background flow. Agents with larger $R/Q$, corresponding to more conservative actuation policies, tend to have smaller actuation velocity and align their actuation in the direction of the flow field to take advantage of the background flow. Except in the most

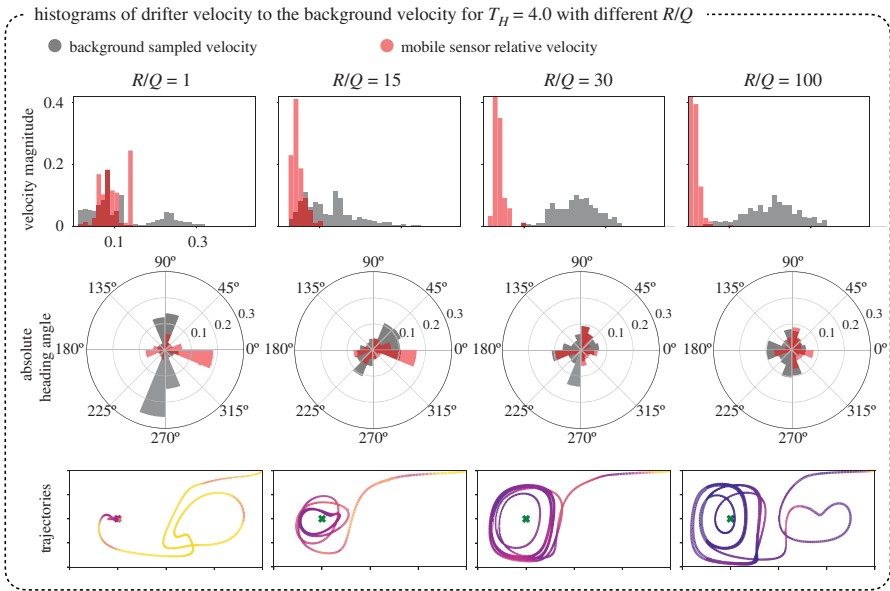

**Figure 8.** As the sensor moves in the double gyre flow field, it is constantly taking control actions $u = [u_x, u_y]$, where $u_x$, $u_y$ are the $x$- and $y$-components of its actuation respectively. At each instant, the sensor is also moving over a background double gyre flow velocity vector whose components, $v_x$ and $v_y$, are given by (3.4). The top row of histograms are of the magnitude of control actions $||u||$ taken (in red), against the magnitude of the background current velocity $||v(x_s, y_s, t)||$ (in grey), where $x_s$, $y_s$ are the sensor coordinates at time $t$. The second row shows the heading angle of the sensor (the orientation of $dx/dt$), plotted in red, against the orientation of the background flow field velocity vector, plotted in grey. The corresponding trajectories are shown on the bottom. (Online version in colour.)

aggressive $R/Q = 1$ case, the mobile sensor rarely uses the maximum control velocity. Additional plots with the $x$- and $y$-components of the agent velocity are presented in figure 16 in appendix A.

## (f) Different start and end locations

Figure 9 shows the MPC optimized trajectories for six different starting locations along the right and lower boundaries of the domain. Although the paths have different initial transients, the trajectories evolve onto the same periodic loitering orbits around the goal state, indicating that they are ultimately leveraging similar flow structures. Similarly, figure 10 shows the MPC optimized trajectories for several different goal states. Some goal states are much more difficult to reach than others, because of the strong unsteady background flow field. With more aggressive control ($R/Q = 1$) the trajectories generally form tighter loitering orbits. However, for the case when the goal state is in the upper middle of the domain, it is clear that none of the MPC trajectories are able to find a suitable loitering pattern. This diversity of orbits highlights the importance of choosing a suitable goal location, which would likely involve a higher level of planning.

## 5. Advanced test cases

From the analyses in the double gyre flow field, we observed the strong dependency of energy-efficient MPC trajectories on the LCS of unsteady periodic flow fields. In this section, we study the use of MPC on more challenging test cases: an analytical three-dimensional incompressible flow field and a real-world flow field reconstructed from ocean model datasets. The goal is to

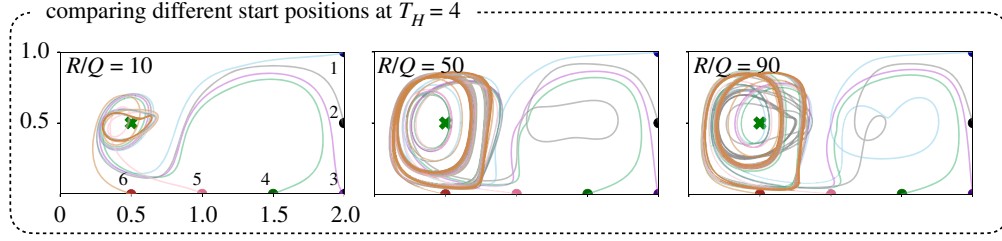

**Figure 9.** MPC trajectories with varying $R/Q$ ratio, starting from different initial conditions with the same goal location. We find that eventually, the trajectories converge to similar periodic orbits thereby using the LCS in similar ways to orbit around the goal location despite having different transients. (Online version in colour.)

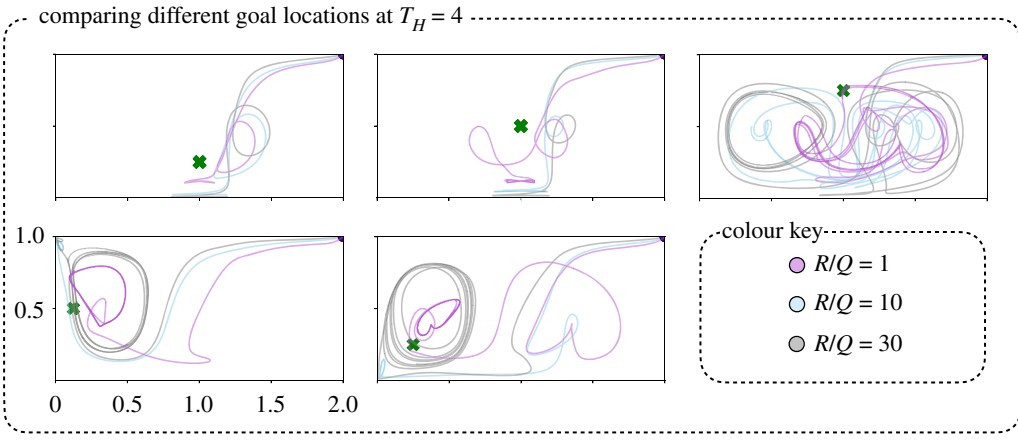

**Figure 10.** MPC trajectories to different goal locations with varying $R/Q$ ratio for the same initial condition. We find that in the case of placing the goal near the middle to bottom region (top left and top middle plots) of double gyre, the sensor is able to form stable orbits near the attracting LCS, where the sensor moves with the base of the attracing LCS. However, placing the goal near the repelling LCS causes more difficulty for MPC in forming small stable orbits (top right). The bottom plots show that it is possible to form small periodic orbits in the corners of the double gyre flow field when the goal is placed close to them. (Online version in colour.)

demonstrate how and when these results generalize to better understand the dependency of energy-efficient MPC trajectories on FTLE-based LCS.

## (a) Arnold–Beltrami–Childress

We now demonstrate the use of MPC on the ABC flow field [79,80], which is an incompressible model for a flow evolving in a three-dimensional periodic domain. It has been studied exhaustively in the past as a stepping stone to understanding turbulent flow fields. The three-dimensional space contains six interwoven vortices. An important feature of this flow field is that even the steady version of the flow field can give rise to chaotic trajectories. However, for our purposes, we investigate the unsteady case. Similar to (3.5) for the double gyre flow field, the equation for a mobile sensor evolving in the ABC flow is given by

$$\frac{\mathrm{d}}{\mathrm{d}t}\begin{bmatrix} x \\ y \\ z \end{bmatrix} = \begin{bmatrix} A(t)\sin(z) + C\cos(y) \\ B\sin(x) + A(t)\cos(z) \\ C\sin(y) + B\cos(x) \end{bmatrix} + \begin{bmatrix} u_x \\ u_y \\ u_z \end{bmatrix}, \tag{5.1}$$

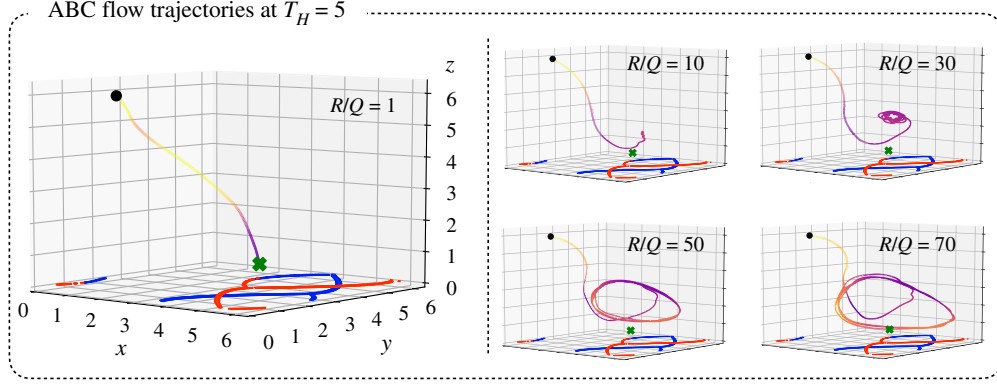

**Figure 11.** MPC trajectories formed by the sensor in an ABC flow field as a function of the $R/Q$ ratio. We observe that for a time horizon of 5, we are able to find several cases of periodic orbits loitering close to the goal location. In these cases, the initial position is $x_{start} = [\pi/2, 1, 6]$ and the goal is $x_{goal} = [5, 2, 1]$. (Online version in colour.)

with parameters $A, B, C, \epsilon, \omega \in \mathbb{R}$, where $A(t) = A + \epsilon \cos(\omega t)$ is a time-varying component that makes the flow field unsteady. We investigate trajectories in the regime where $A : B : C = \sqrt{3} : \sqrt{2} : 1$, which has been exhaustively studied numerically and analytically (with $C = 0.1$ specifically in our simulations), $u_x, u_y, u_z \leq A + B + C$ and $\omega = 2\pi/10$. All ABC simulations presented were run for 2000 time steps with a step size of $\Delta t = 0.1$. The cost function used was the same as equation (3.6). In figure 11, we show the trajectories planned by the MPC for a time horizon $T_H = 5$. Across these simulations, we observed that, similar to the double gyre, it is possible to form loitering orbits close to the goal point. These loitering orbits become larger for larger $R/Q$. Results across different time horizons and $R/Q$ are further summarized in figure 12. We found that relatively short time horizon (compared to the period of oscillation $T = 10$) trajectories are able to reach the goal state, and similar to the double gyre, longer time horizons reduce error with lesser energy consumption. The inflection points and breaking off of points from the curve (for example, the black dots near $E = 1500$) correspond to drastic changes in trajectory shape (bifurcations).

## (b) Gulf of Mexico

For the final example, we consider the Gulf of Mexico surface velocity estimates from the HYbrid Coordinate Ocean Model (HYCOM). This data-assimilative model synthesizes remotely sensed and *in situ* measurements on a hybrid coordinate system. We used daily $1/12.5°$-resolution data from the HYCOM 1992–1995 experiment 19.0 (top left in figure 13) to generate a vector field. We then used linear interpolation on this vector field in space and time to generate a function that could be used for model predictive control. The parameters were chosen to be $\Delta t = 0.1$ day, $T_H = 0.4$ day, $u_x, u_y \leq 2$ km h$^{-1}$, and $R/Q = 1$. The step size $\Delta t = 2.4$ h for the MPC. The full trajectories in figure 14 were computed for 1000 time steps (100 days).

We observe similar spiking behaviour in the energy spent when moving across a repelling LCS, which is seen in panels $t = 6$ to $t = 10$ of figure 13. Here, $t$ is the time in days (specifically, $t = 0$ corresponds to the flow field on Day-1 of the HYCOM dataset). We observe that the sensor synchronizes with the attracting LCS to move towards the goal location. In periodic flow fields such as the double gyre, we observe the formation of periodic orbits. However, in case of the Gulf of Mexico (aperiodic), we observed the formation of an aperiodic loitering trajectory in proximity to the goal location. The time horizon here is relatively short compared to the total trajectory time. We can see that even with short time horizons the MPC trajectories are capable of making it to the goal location. Although the trajectory does evolve towards a loitering orbit near the goal state, it

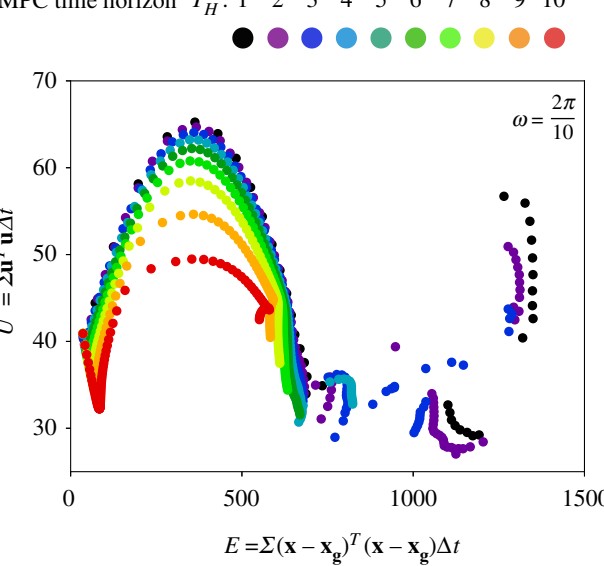

**Figure 12.** This scatter plot shows the performance of MPC in the ABC flow field similar to figure 7. We observe that MPC trajectories with short time horizons are able to reach the goal state and increasing the time horizon has a benefit of decreasing the actuation energy usage. We also observe that the inflection points and breaking off points correspond to bifurcations in the trajectories. (Online version in colour.)

can be seen in figure 14 that for more aggressive control ($R/Q = 0.5$ and $R/Q = 0.1$), the trajectory is able to get much closer to the goal state, with a tighter orbit.

## 6. Discussion and conclusion

In this work, we have investigated the behaviour of trajectories optimized over a finite horizon for a controlled mobile sensor in unsteady flow fields, as both the control and flow field parameters were varied. A thorough study was conducted on the double gyre flow field, and selected results were further verified on more advanced flow fields such as the ABC and the Gulf of Mexico. In particular, finite-time MPC was used to generate energy-efficient trajectories for a range of parameters, particularly the prediction horizon and the relative penalty between the state error and control effort. The double gyre oscillation frequency was varied to study its influence on the resulting trajectories. We have constrained the maximum actuation velocity to be less than the largest background flow velocity such that some degree of intelligent planning is required to efficiently traverse the flow.

Through a quantitatively exhaustive study, we have uncovered several interesting trends and established connections between the finite-horizon optimized mobile sensor trajectories and the coherent structures of the underlying flow field. By varying the relative cost of actuation and deviations in the state (i.e. $R/Q$), the control cost and state error sweep out a Pareto front, and there is often a *sweet spot* where relatively good state tracking performance can be achieved with low actuation costs. These energy-efficient trajectories tend to align with the LCS to take advantage of the unsteady background flow. We also found that locations where the sensor spends most energy correlated with the presence of repelling hyperbolic LCS. These findings could be better understood by recalling that repelling LCS are defined as material barriers in the flow field. This explains the expenditure of more energy close to the repelling LCS, as energy needs to be spent to overcome barrier for movement. Furthermore, as the energy spent for movement goes to zero, the sensor behaves similar to a passive tracer. It is known that the short-term dynamics of passive

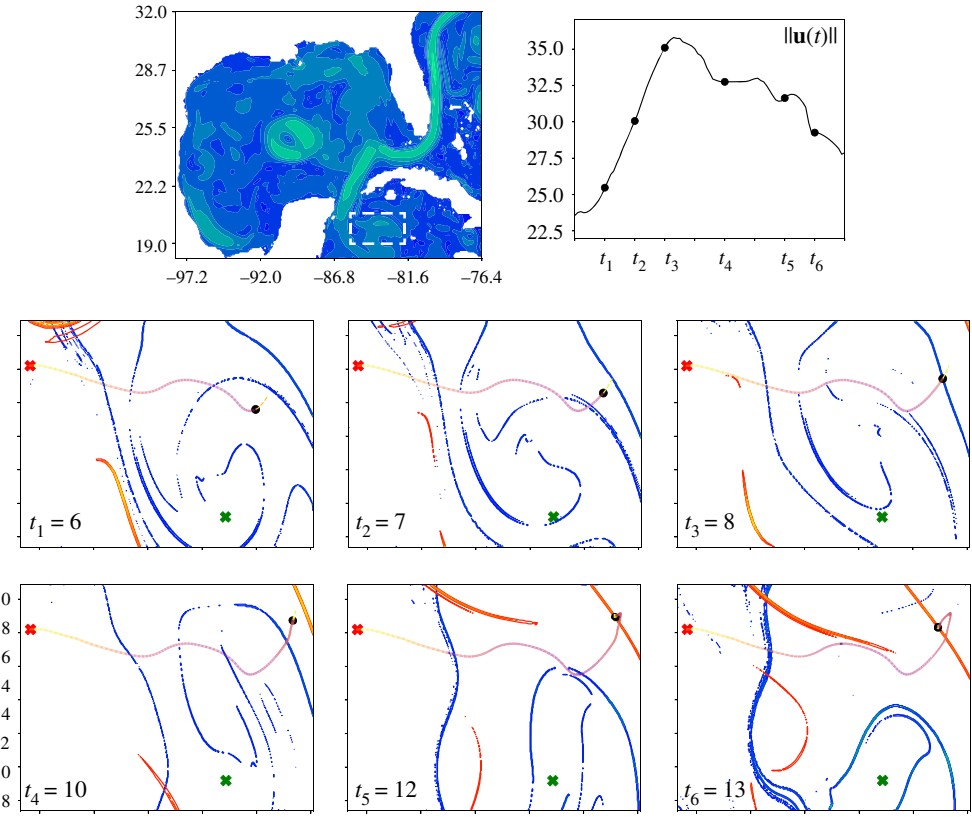

**Figure 13.** This figure shows the use of MPC to plan trajectories in the Gulf of Mexico dataset. We have chosen the region highlighted in dashed-dotted lines in the top left plot. The bottom six plots show the trajectory generated for $R/Q = 1$ (full trajectory can be seen in the rightmost plot in figure 14) and $u_x, u_y \leq 2$ km h$^{-1}$ in colour shading from yellow to purple to highlight how energy is spent along the path. The units on the $x$- and $y$-axis are longitude and latitude, respectively. The red cross shows start location, $x_{\text{start}} = [-85.5, 19.8]$, the green cross shows the goal location, $x_{\text{goal}} = [-83.7, 18.9]$. The black dot shows the instantaneous sensor location. Viewing the six panels in sequence we observe that the sensor moves across the blue repelling LCS from $t = 6$ to $t = 8$, where $t$ is the time in days. We observe a spike in the instantaneous energy spent and a slow drop as we move away from the repelling LCS (as seen in the top right plot of $u(t)$). We then observe that the sensor synchronizes with the attracting LCS to move towards the goal location. (Online version in colour.)

tracers are governed by movement with the FTLE. This suggests that these connections hold for energy optimality. Importantly, we find that it is often possible to generate effective, energy-efficient mobile sensor trajectories with a relatively short prediction horizon, which is promising for the future design of trajectories with limited or partial knowledge of the background flow field.

We observe a rough trend of lower state error when control is less expensive, which agrees with the intuition that the agent is able to more directly pursue the goal state by actuating more aggressively. However, this trend is not monotonic, as there are several cases where slightly decreasing the control cost results in worse state tracking performance. These non-monotonic changes in the cost versus $R/Q$ correspond to *bifurcations* in the agent trajectories, which either correspond to longer trajectories, or to discontinuous jumps in the shape and size of the periodic orbit around the goal state. These bifurcations are more common for smaller prediction horizons, which is also consistent with the intuition that smaller prediction horizons may lead the agent to get trapped by unfavourable flow structures. Similarly, for a fixed relative control cost, there are

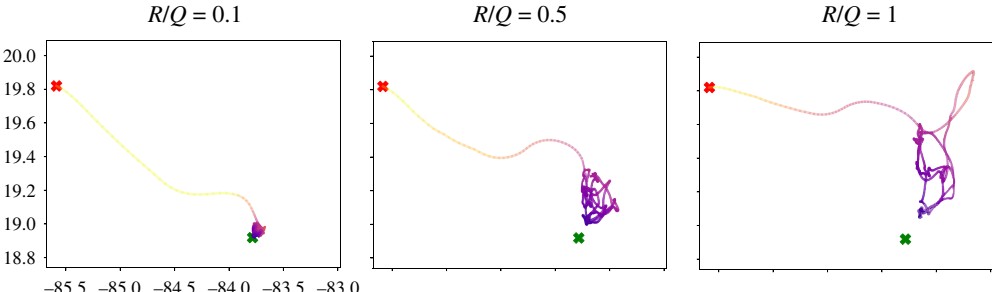

**Figure 14.** This figure shows the change in MPC trajectories in the Gulf of Mexico as the $R/Q$ ratio is varied. We see an aperiodic loitering state near the goal location. Given that the time horizon in this case $T_H = 9.6$ h, which is relatively short compared to the total trajectory time of 100 days, we can see that even with short time horizons the MPC trajectories are capable of making it to the goal location. (Online version in colour.)

bifurcations in the optimized trajectory with variations in the time horizon. These bifurcations are relevant in the context of generating mobile sensor trajectories using MPC, as small changes in the weights can lead the drastically different trajectories. Upon closer inspection, these bifurcations correspond to the agent trajectory passing through a Lagrangian coherent structure, after which the two trajectory behaviours diverge.

It is also important to note that the energy-efficient trajectories typically result in periodic orbits around the target position, since the unsteady double gyre is periodically oscillating. Previous studies in trajectory generation have mainly focused on solving boundary value optimizations for trajectories keeping the start and end points fixed. Our results show that these assumptions can be relaxed, and moreover, it is possible to reach periodic steady states with little actuation even when the uncontrolled drifter dynamics are chaotic. These periodic orbits correspond to, often desirable, station keeping or hovering behaviour.

This work has several implications for the control of individual mobile robots and swarms of robots in geophysical flows. The ability to generate energy-efficient trajectories that take advantage of the background flow with a short prediction horizon is promising for practical applications. The ability to maintain close periodic orbits around the goal state may also enable efficient long-time monitoring. For example, fix-wing unmanned aerial vehicles must often loiter over an area for sensing and monitoring. Variations in the shape of periodic orbits and the Pareto optimal curves over different $R/Q$ ratios with the gyre oscillation frequency have implications for ocean applications, which exhibit a wide range of spatio-temporal scales with varying oscillating frequencies. We also observed an increase in the expenditure of the sensor's actuation energy as it approached background LCS. This result is beneficial in the context of identifying background coherent structures by observing the energy expenditure patterns of controlled agents. This is an important problem with ongoing work [81–83]. These results are also potentially useful in the design of scalable navigation algorithms for mobile sensor swarms where the objective is to maintain cohesion or connectivity between agents.

This work motivates a number of interesting future directions. Our results indicate that it is possible to design nearly optimal, energy-efficient trajectories, even with short prediction horizons for the MPC; however, it was assumed that the background flow was known perfectly for this short horizon. It will be important to further explore the robustness of these trajectory optimizations to more realistic scenarios with partial, noisy and uncertain information about the background flow. This analysis may benefit from recent works that have investigated the sensitivity of FTLE calculations to uncertain flow field data [84,85] as well as how FTLE can be used to propagate uncertainties through chaotic flow maps [86]. Because the optimization result depends strongly on how the MPC trajectories interact with LCS of the background flow field, it may also be possible to incorporate knowledge about the LCS more directly to

the optimization. Even with uncertain or partially observed flow field information, often the LCS are quite persistent, and it may be possible to develop time-varying *maps* of the coherent structures in different geographical regions, for example off the Horn of Africa or in the Gulf of Mexico. In addition, it will be interesting to explore the use of other coherent structure and modal decomposition identification techniques [87,88]. Further study is also required to characterize the dynamics and coherent structures of the *controlled* vector field of the agent given a specific control policy. In addition, all the results in this paper were developed through the study of the double gyre flow field. It will be interesting to perform similar investigations for a variety of flow fields. For example, it will be important to explore how these results change when the flow exhibits a wider range of multiscale behaviour in space and time. Particularly, multiscale turbulence will affect the prediction horizon, as uncertainties will be magnified making it challenging to forecast flow structures. These multiscale structures will also impact energy efficiency, both through the forecast uncertainty, but also through making optimal paths more circuitous. Finally, extending the analysis to additional *three-dimensional* turbulent flows will also be critical.

Data accessibility. The code used in this study is openly available at https://github.com/karkris41295/single-agent-MPC-FTLE.

Authors' contributions. All authors conceived of the work. K.K. performed computations and generated data and figures. All authors analysed data. All authors wrote the manuscript and contributed to its revision. All authors gave final approval for publication and agree to be held accountable for the work performed therein.

Competing interests. We declare we have no competing interests.

Funding. S.L.B. acknowledges funding support from the Air Force Office of Scientific Research (AFOSR FA9550-18-1-0200) and the Army Research Office (ARO W911NF-19-1-0045). Z.S. acknowledges funding support from the National Science Foundation (NSF IIS-2024928 and OIA-2032522).

## Appendix A

Here, we present additional information that provides a more detailed analysis of the performance of MPC trajectories for various parameters. In addition to these extra figures, we point the reader to the online videos.

In figure 15, we see the evolution of trajectories with $R/Q = 2$ and $R/Q = 3$ with a time horizon $T_H = 4$ to explain why the $R/Q = 2$ trajectory initially appears to perform worse than the $R/Q = 3$ trajectory in figure 2 from the main text. In particular, it appears that the more aggressive $R/Q = 2$ agent ends up on the wrong side of the blue LCS, which forces it to take a full revolution in the right gyre before making it to the left gyre where the goal state resides. We observe these phenomena in several different parameter regimes, where small changes in the parameters may cause agents to get forced into extra orbits in the right gyre.

Figures S1–S9 in the electronic supplementary material provide similar information to figure 2 in the main text, but with different time horizons. Even for a short time horizon of $T_H = 1$, the most aggressive controllers achieve relatively good state tracking performance. However, the cost versus $R/Q$ curves for $T_H \leq 3$ are considerably less monotonic than those for $T_H \geq 4$, indicating several more bifurcations in the trajectory shape. For $T_H \in [4, 7]$, the behaviour is fairly regular, exhibiting the same qualitative bifurcation behaviour. Interestingly, there is a trend of bifurcations occurring later for larger $T_H$ in this range, as the longer time-horizon controllers are able to achieve slightly better trajectories for larger $R/Q$ values.

Finally, figure 16 provides the histograms of the *x*- and *y*-components of the agent velocity, complementing the data in figure 8 from the main text.

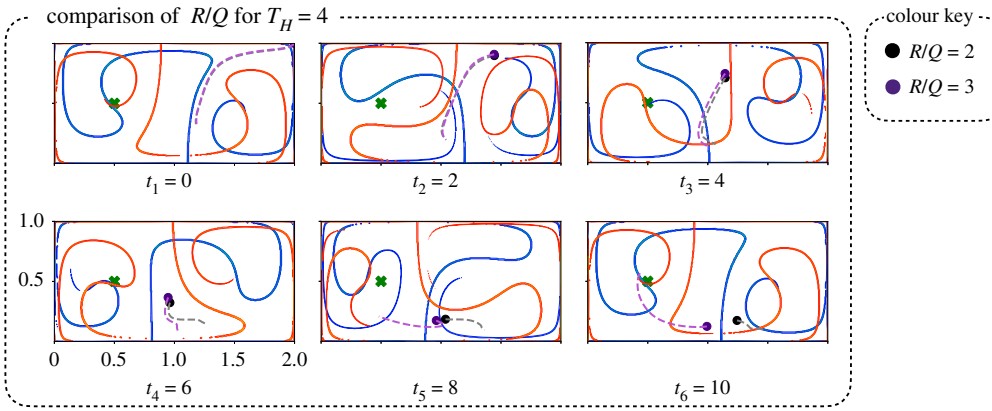

**Figure 15.** In figure 2, we observe that for $T_H = 4$, $R/Q = 2$, the trajectory makes a loop in the right gyre before ultimately reaching the goal. This is not the case for $R/Q = 3$ which is more greedy in spending energy. This is somewhat counterintuitive, since we expect that spending more energy should drive the mobile sensor more rapidly to the goal. In this plot, we show the difference in how the control energy is spent comparing $R/Q = 2$ and $R/Q = 3$. We observe that $R/Q = 2$ takes longer to reach the goal because good performance is very sensitive to timing in the gyre. We see that the aggressive control (grey trajectory) moves too far ahead of the non-aggressive control (purple trajectory) to make use of the gyre dynamics to move directly to the goal state. (Online version in colour.)

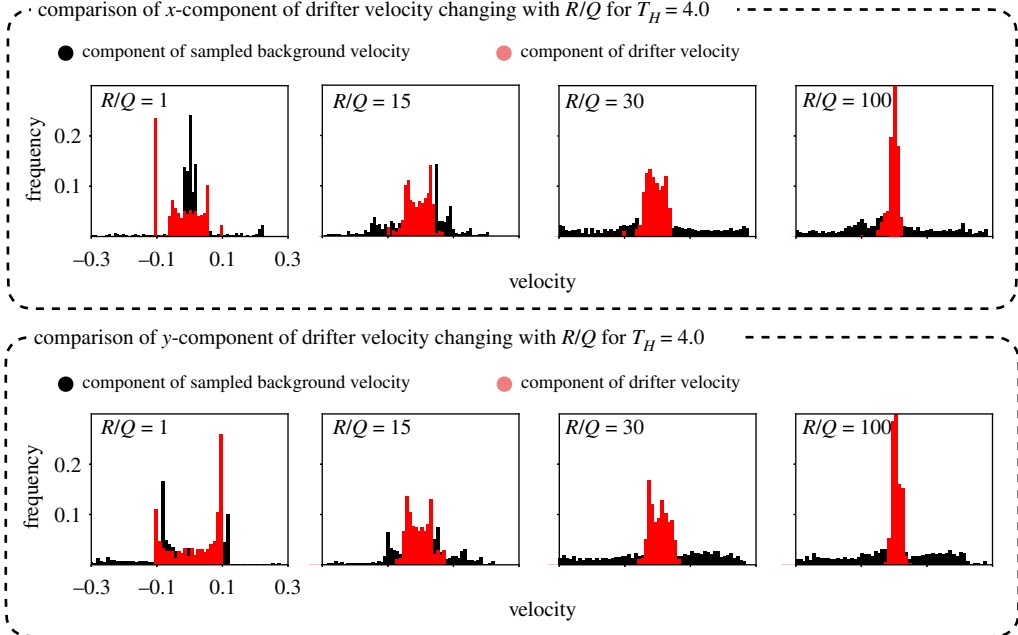

**Figure 16.** As the sensor moves in the double gyre flow field, it is constantly taking control actions $u = [u_x, u_y]$, where $u_x$, $u_y$ are the $x$- and $y$-components of its actuation, respectively. At each instant, the sensor is also moving over a background double gyre current vector whose components $v_x$, $v_y$ are given by (3.4). The top row is a histogram of the $x$-component of control actions $u_x$ taken (in red), against the $x$-component of the background current velocity $v_x(x_s, y_s, t)$ (in black), where $x_s$, $y_s$ are the sensor coordinates at time $t$. The second row is a similar plot for the $y$-component. We observe that the gyre takes values beyond the actuation capacity of the sensor, which highlights the under-actuated nature of the problem. Also, at low $R/Q$ ratios, the distribution of control actions follows a distribution with two peaks at $\pm 0.1$, which corresponds to a situation similar to bang-bang control. As we increase the $R/Q$ ratio, the distribution of control actions moves to a single peak centred around zero corresponding to the use of very little control effort when compared with the background velocity. (Online version in colour.)

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
