## [Peer Review File · Proceedings. Mathematical, Physical, and Engineering Sciences]

Review History

RSPA-2021-0255.R0 (Original submission)

Review form: Referee 1

Is the manuscript an original and important contribution to its field?

Acceptable

Is the paper of sufficient general interest?

Good

Is the overall quality of the paper suitable?

Excellent

Can the paper be shortened without overall detriment to the main message?

Yes

Do you think some of the material would be more appropriate as an electronic appendix?

No

Do you have any ethical concerns with this paper?

No

Recommendation?

Major revision is needed (please make suggestions in comments)

Comments to the Author(s)

The manuscript is concerned with using model predictive control to obtain energy-efficient trajectories for vehicles which are subject to spatially and temporally varying background flows (wind, ocean currents, etc). Key aspects of the manuscript are the use of very short time horizons and the observation for the test case that the resulting efficient trajectories are meaningfully exploiting structures in the background flow.

Overall the manuscript is clear and easy to follow. However, the conclusions are weakened somewhat by the consideration of a single 2D test case where it is assumed that the background flow field is known perfectly and that the vehicle does not meaningfully influence field. Whilst these set of assumptions are reasonable for ocean vehicles, they are likely to be deficient for any aerial applications (several of which are mentioned in the abstract, introduction, and conclusion).

The manuscript would therefore benefit greatly from a 3D test case (which should be a straightforward extension) with a more erratic background field. Or, alternatively, an attempt to incorporate measurement noise into the background field. Having one, or ideally both of these, would go a long way towards enhancing the impact of the manuscript.

Specific remarks:

- p5 Eq (2.3) is awkward to unpick and would benefit from introducing a grid-spacing variable h_{ij} .
- p6 Wording: 'More broadly, FTLE has been used to coherent structures'

Review form: Referee 2

Is the manuscript an original and important contribution to its field?

Marginal

Is the paper of sufficient general interest?

Marginal

Is the overall quality of the paper suitable?

Acceptable

Can the paper be shortened without overall detriment to the main message?

Yes

Do you think some of the material would be more appropriate as an electronic appendix?

Yes

Do you have any ethical concerns with this paper?

No

Recommendation?

Major revision is needed (please make suggestions in comments)

Comments to the Author(s)

Inspired by the application of mobile sensors, the paper centers around the application of finite horizon model predictive control (MPC) and finite time Lyapunov exponents (FTLE) using simple test problems and synthetic flow fields. The key premise of the paper is to find mobile sensor trajectories for a relatively short horizon using limited or partial knowledge of the

background flow field. The authors attempt to understand the sensitivity of the trajectories and estimate an optimal range where energy-efficient trajectories are found. Overall the paper is well written and has some scientific relevance on unsteady flow prediction & control of mobile sensors. My main criticism is with regard to the novelty and practical relevance of the presented results. Some specific concerns are as follows:

1. I'm not clear about the novelty and archival value of this paper. I want the authors to articulate their specific contributions and their practical demonstration. MPC and FTLE are standard approaches in control theory and dynamical systems. The codes and libraries are easily available. The results are based on toy problems and they lack generality to real world situation.
2. In Section 3, why the flow field is considered simple, instead of real flow field from CFD or experimental data? The current results have no relevance to any practical problem. Unfortunately, the presented results lack physical interpretation. They are several trajectories presented but it's not clear why and how? Also there are no verification and validation of the claims the authors have made about the prediction horizon and the energy optimal trajectories.
3. While the ocean flow is turbulent, the authors should discuss the impact of turbulence into their prediction horizons and the energy efficiency.

Review form: Referee 3

Is the manuscript an original and important contribution to its field?

Marginal

Is the paper of sufficient general interest?

Acceptable

Is the overall quality of the paper suitable?

Marginal

Can the paper be shortened without overall detriment to the main message?

Yes

Do you think some of the material would be more appropriate as an electronic appendix?

No

Do you have any ethical concerns with this paper?

No

Recommendation?

Major revision is needed (please make suggestions in comments)

Comments to the Author(s)

Please see attached file.

Decision letter (RSPA-2021-0255.R0)

21-Jul-2021

Dear Mr Krishna

The Editor of Proceedings A has now received comments from referees on the above paper and would like you to revise it in accordance with their suggestions which can be found below (not including confidential reports to the Editor).

Please submit a copy of your revised paper within four weeks - if we do not hear from you within this time then it will be assumed that the paper has been withdrawn. In exceptional circumstances, extensions may be possible if agreed with the Editorial Office in advance.

Please note that it is the editorial policy of Proceedings A to offer authors one round of revision in which to address changes requested by referees. If the revisions are not considered satisfactory by the Editor, then the paper will be rejected, and not considered further for publication by the journal. In the event that the author chooses not to address a referee's comments, and no scientific justification is included in their cover letter for this omission, it is at the discretion of the Editor whether to continue considering the manuscript.

To revise your manuscript, log into <http://mc.manuscriptcentral.com/prsa> and enter your Author Centre, where you will find your manuscript title listed under "Manuscripts with Decisions." Under "Actions," click on "Create a Revision." Your manuscript number has been appended to denote a revision.

You will be unable to make your revisions on the originally submitted version of the manuscript. Instead, revise your manuscript and upload a new version through your Author Centre.

When submitting your revised manuscript, you will be able to respond to the comments made by the referee(s) and upload a file "Response to Referees" in Step 1: "View and Respond to Decision Letter". Please provide a point-by-point response to the comments raised by the reviewers and the editor(s). A thorough response to these points will help us to assess your revision quickly. You can also upload a 'tracked changes' version either as part of the 'Response to reviews' or as a 'Main document'.

IMPORTANT: Your original files are available to you when you upload your revised manuscript. Please delete any unnecessary previous files before uploading your revised version.

When revising your paper please ensure that it remains under 28 pages long. In addition, any pages over 20 will be subject to a charge (£150 + VAT (where applicable) per page). Your paper has been ESTIMATED to be 24 pages.

Open Access

You are invited to opt for open access, our author pays publishing model. Payment of open access fees will enable your article to be made freely available via the Royal Society website as soon as it is ready for publication. For more information about open access please visit <https://royalsociety.org/journals/authors/open-access/>. The open access fee for this journal is £1700/\$2380/€2040 per article. VAT will be charged where applicable. Please note that if the corresponding author is at an institution that is part of a Read and Publishing deal you are required to select this option. See <https://royalsociety.org/journals/librarians/purchasing/read-and-publish/read-publish-agreements/> for further details.

Once again, thank you for submitting your manuscript to Proc. R. Soc. A and I look forward to receiving your revision. If you have any questions at all, please do not hesitate to get in touch.

Yours sincerely
Raminder Shergill
proceedingsa@royalsociety.org

on behalf of
 Professor Graham Hughes
 Board Member
 Proceedings A

Reviewer(s)' Comments to Author:

Referee: 1

Comments to the Author(s)

The manuscript is concerned with using model predictive control to obtain energy-efficient trajectories for vehicles which are subject to spatially and temporally varying background flows (wind, ocean currents, etc). Key aspects of the manuscript are the use of very short time horizons and the observation for the test case that the resulting efficient trajectories are meaningfully exploiting structures in the background flow.

Overall the manuscript is clear and easy to follow. However, the conclusions are weakened somewhat by the consideration of a single 2D test case where it is assumed that the background flow field is known perfectly and that the vehicle does not meaningfully influence field. Whilst these set of assumptions are reasonable for ocean vehicles, they are likely to be deficient for any aerial applications (several of which are mentioned in the abstract, introduction, and conclusion).

The manuscript would therefore benefit greatly from a 3D test case (which should be a straightforward extension) with a more erratic background field. Or, alternatively, an attempt to incorporate measurement noise into the background field. Having one, or ideally both of these, would go a long way towards enhancing the impact of the manuscript.

Specific remarks:

- p5 Eq (2.3) is awkward to unpick and would benefit from introducing a grid-spacing variable h_{ij} .
- p6 Wording: 'More broadly, FTLE has been used to coherent structures'

Referee: 2

Comments to the Author(s)

Inspired by the application of mobile sensors, the paper centers around the application of finite horizon model predictive control (MPC) and finite time Lyapunov exponents (FTLE) using simple test problems and synthetic flow fields. The key premise of the paper is to find mobile sensor trajectories for a relatively short horizon using limited or partial knowledge of the background flow field. The authors attempt to understand the sensitivity of the trajectories and estimate an optimal range where energy-efficient trajectories are found. Overall the paper is well written and has some scientific relevance on unsteady flow prediction & control of mobile sensors. My main criticism is with regard to the novelty and practical relevance of the presented results. Some specific concerns are as follows:

1. I'm not clear about the novelty and archival value of this paper. I want the authors to articulate their specific contributions and their practical demonstration. MPC and FTLE are standard approaches in control theory and dynamical systems. The codes and libraries are easily available. The results are based on toy problems and they lack generality to real world situation.
2. In Section 3, why the flow field is considered simple, instead of real flow field from CFD or experimental data? The current results have no relevance to any practical problem. Unfortunately, the presented results lack physical interpretation. They are several trajectories presented but it's not clear why and how? Also there are no verification and validation of the claims the authors have made about the prediction horizon and the energy optimal trajectories.
3. While the ocean flow is turbulent, the authors should discuss the impact of turbulence into their prediction horizons and the energy efficiency.

Referee: 3

Comments to the Author(s)

Please see attached file.

Board Member:

Comments to Author(s):

The reviewers are all supportive of your paper and have made a number of constructive comments to be addressed in producing a revised version. An overriding theme is the request to demonstrate and discuss the application of your techniques to a more complicated or realistic flow. Please consider how this outcome might be best achieved and return a point-by-point response to the reviewers' comments with your revised manuscript.

Board member pre-assessment comments (if available):

This paper is on an interesting topic, has many potential applications and is written to be accessible to a wide range of PRSA readers. There look to be many exciting avenues for future work in the area.

Author's Response to Decision Letter for (RSPA-2021-0255.R0)

See Appendix A.

RSPA-2021-0255.R1 (Revision)

Review form: Referee 1

Is the manuscript an original and important contribution to its field?

Good

Is the paper of sufficient general interest?

Good

Is the overall quality of the paper suitable?

Excellent

Can the paper be shortened without overall detriment to the main message?

Yes

Do you think some of the material would be more appropriate as an electronic appendix?

No

Do you have any ethical concerns with this paper?

No

Recommendation?

Accept as is

Comments to the Author(s)

I would like to thank the reviewers for their detailed responses to my concerns and for suitably revising the manuscript.

Review form: Referee 2

Is the manuscript an original and important contribution to its field?

Acceptable

Is the paper of sufficient general interest?

Acceptable

Is the overall quality of the paper suitable?

Acceptable

Can the paper be shortened without overall detriment to the main message?

Yes

Do you think some of the material would be more appropriate as an electronic appendix?

Yes

Do you have any ethical concerns with this paper?

No

Recommendation?

Accept as is

Comments to the Author(s)

Thank you for incorporating my comments.

Decision letter (RSPA-2021-0255.R1)

29-Nov-2021

Dear Mr Krishna

I am pleased to inform you that your manuscript entitled "Finite-Horizon, Energy-Efficient Trajectories in Unsteady Flows" has been accepted in its final form for publication in Proceedings A.

Our Production Office will be in contact with you in due course. You can expect to receive a proof of your article soon. Please contact the office to let us know if you are likely to be away from e-mail in the near future. If you do not notify us and comments are not received within 5 days of sending the proof, we may publish the paper as it stands.

As a reminder, you have provided the following 'Data accessibility statement' (if applicable). Please remember to make any data sets live prior to publication, and update any links as needed when you receive a proof to check. It is good practice to also add data sets to your reference list. Statement (if applicable): The code used to generate the data in the paper is available at <https://github.com/karkris41295/single-agent-MPC-FTLE>

Under the terms of our licence to publish you may post the author generated postprint (ie. your accepted version not the final typeset version) of your manuscript at any time and this can be made freely available. Postprints can be deposited on a personal or institutional website, or a recognised server/repository. Please note however, that the reporting of postprints is subject to a

media embargo, and that the status the manuscript should be made clear. Upon publication of the definitive version on the publisher's site, full details and a link should be added.

You can cite the article in advance of publication using its DOI. The DOI will take the form: 10.1098/rspa.XXXX.YYYY, where XXXX and YYYY are the last 8 digits of your manuscript number (eg. if your manuscript number is RSPA-2017-1234 the DOI would be 10.1098/rspa.2017.1234).

For tips on promoting your accepted paper see our blog post:
<https://royalsociety.org/blog/2020/07/promoting-your-latest-paper-and-tracking-your-results/>

On behalf of the Editor of Proceedings A, we look forward to your continued contributions to the Journal.

Sincerely,
Raminder Shergill
proceedingsa@royalsociety.org

on behalf of
Professor Graham Hughes
Board Member
Proceedings A

Reviewer(s)' Comments to Author:

Referee: 1

Comments to the Author(s)

I would like to thank the reviewers for their detailed responses to my concerns and for suitably revising the manuscript.

Referee: 2

Comments to the Author(s)

Thank you for incorporating my comments.

Appendix A

Response to referee comments on “FINITE HORIZON ENERGY OPTIMAL TRAJECTORIES IN UNSTEADY FLOWS”

Dear Referees,

We are grateful for your careful reviews and for the insightful suggestions. These comments have provided us with valuable perspectives and have helped us improve our arguments considerably. We have attempted to address each of the reviewer comments throughout the manuscript. Below is a detailed summary of how each review was addressed. Responses are color coded here as blue and in the text as red.

All referees shared the concern that only exploring the double gyre flow field was insufficient for several of the claims in the manuscript. In the revised manuscript, we have added two new more challenging test flows: a 3D chaotic flow field (the ABC flow), and data from the Gulf of Mexico. These results are presented in the new Section 5, and corroborate many of the findings with the double gyre. We appreciate the referees pushing us to explore these more interesting and challenging test cases, and we believe the revised manuscript has been strengthened considerably.

The results of these new test cases are summarized here, instead of in the individual responses below, as they address all three referee comments.

Advanced Test Cases

From the analyses in the double gyre flow field, we observed the strong dependency of energy-efficient MPC trajectories on the LCS of unsteady periodic flow fields. In this section, we study the use of MPC on more challenging test cases: an analytical three-dimensional incompressible flow field and a real-world flow field reconstructed from ocean model datasets. The goal is to demonstrate how and when these results generalize to better understand the dependency of energy-efficient MPC trajectories on FTLE-based LCS.

Arnold-Beltrami-Childress

We now demonstrate the use of MPC on the ABC flow field [79, 80], which is an incompressible model for a flow evolving in a three-dimensional periodic domain. It has been studied exhaustively in the past as a stepping stone to understanding turbulent flow fields. The three-dimensional space contains six interwoven vortices. An important feature of this flow field is that even the steady version of the flow field can give rise to chaotic trajectories.

Figure 11: MPC trajectories formed by the sensor in an ABC flow field as a function of the R/Q ratio. We observe that for a time horizon of 5, we are able to find several cases of periodic orbits loitering close to the goal location. In these cases, the initial position is $\mathbf{x}_{\text{start}} = [\pi/2, 1, 6]$ and the goal is $\mathbf{x}_{\text{goal}} = [5, 2, 1]$.

However, for our purposes, we investigate the unsteady case. Similar to (??) for the double gyre flow field, the equation for a mobile sensor evolving in the ABC flow is given by

$$\frac{d}{dt} \begin{bmatrix} x \\ y \\ z \end{bmatrix} = \begin{bmatrix} A(t) \sin(z) + C \cos(y) \\ B \sin(x) + A(t) \cos(z) \\ C \sin(y) + B \cos(x) \end{bmatrix} + \begin{bmatrix} u_x \\ u_y \\ u_z \end{bmatrix} \quad (1)$$

with parameters $A, B, C, \epsilon, \omega \in \mathbb{R}$, where $A(t) = A + \epsilon \cos(\omega t)$ is a time-varying component that makes the flow field unsteady. We investigate trajectories in the regime where $A : B : C = \sqrt{3} : \sqrt{2} : 1$, which has been exhaustively studied numerically and analytically (with $C = 0.1$ specifically in our simulations), $u_x, u_y, u_z \leq A + B + C$ and $\omega = 2\pi/10$. All ABC simulations presented were run for 2000 time steps with a step size of $\Delta t = 0.1$. The cost function used was the same as Eq. 3.6. In Figure 11, we show the trajectories planned by the MPC for a time horizon $T_H = 5$. Across these simulations, we observed that, similar to the double gyre, that it is possible to form loitering orbits close to the goal point. These loitering orbits become larger for larger R/Q . Results across different time horizons and R/Q are further summarized in Figure 12. We found that relatively short time horizon (compared to the period of oscillation $T = 10$) trajectories are able to reach the goal state, and similar to the double gyre, longer time horizons reduce error with lesser energy consumption. The inflection points and breaking off of points from the curve (for example, the black dots near $E = 1500$) correspond to drastic changes in trajectory shape (bifurcations).

Gulf of Mexico

For the final example, we consider the Gulf of Mexico surface velocity estimates from the

Figure 12: This scatter plot shows the performance of MPC in the ABC flow field similar to Figure 7. We observe that MPC trajectories with short time horizons are able to reach the goal state and increasing the time horizon has a benefit of decreasing the actuation energy usage. We also observe that the inflection points and breaking off points correspond to bifurcations in the trajectories.

HYbrid Coordinate Ocean Model (HYCOM). This data-assimilative model synthesizes remotely sensed and in-situ measurements on a hybrid coordinate system. We used daily $1/12.5^\circ$ -resolution data from the HYCOM 1992-1995 experiment 19.0 to generate a vector field. We then used linear interpolation on this vector field in space and time to generate a function that could be used for model predictive control. The parameters were chosen to be $\Delta t = 0.1$ day, $T_H = 0.4$ day, $u_x, u_y \leq 2$ km/hr, and $R/Q = 1$. The step size $\Delta t = 2.4$ hours for the MPC. The full trajectories in Figure 14 were computed for 1000 time steps (100 days).

We observe similar spiking behaviour in the energy spent when moving across a repelling LCS, which is seen in panels $t = 6$ to $t = 10$. Here, t is the time in days (specifically, $t = 0$ corresponds to the flow field on Day-1 of the HYCOM dataset). We observe that the sensor synchronizes with the attracting LCS to move towards the goal location. In periodic flow fields such as the double gyre, we observe the formation of periodic orbits. However, in case of the Gulf of Mexico (aperiodic), we observed the formation of an aperiodic loitering trajectory in proximity to the goal location. The time horizon here is relatively short compared to the total trajectory time. We can see that even with short time horizons the MPC trajectories are capable of making it to the goal location. Although the trajectory does evolve towards a loitering orbit near the goal state, it can be seen in Figure 14 that for more aggressive control ($R/Q = 0.5$ and $R/Q = 0.1$), the trajectory is able to get much closer to the goal state, with a tighter orbit.

Figure 13: This figure shows the use of MPC to plan trajectories in the Gulf of Mexico dataset. We have chosen the region highlighted in dashed dotted lines in the top left plot. The bottom six plots show the trajectory generated for $R/Q = 1$ (full trajectory can be seen in the rightmost plot in Figure 14) and $u_x, u_y \leq 2$ km/hr in color shading from yellow to purple to highlight how energy is spent along the path. The units on the x and y axis are longitude and latitude respectively. The red cross in the figures show start location, $\mathbf{x}_{\text{start}} = [-85.5, 19.8]$, the green cross shows the goal location, $\mathbf{x}_{\text{goal}} = [-83.7, 18.9]$. The black dot shows the instantaneous sensor location. Viewing the six figures in sequence we observe that the sensor moves across the blue repelling LCS from $t = 6$ to $t = 8$, where t is the time in days. We observe a spike in the instantaneous energy spent and a slow drop as we move away from the repelling LCS (as seen in the top right plot of $u(t)$). We then observe that the sensor synchronizes with the attracting LCS to move towards the goal location.

Figure 14: This figure shows the change in MPC trajectories in the Gulf of Mexico as the R/Q ratio is varied. We see an aperiodic loitering state near the goal location. Given that the time horizon in this case $T_H = 9.6$ hours, which is relatively short compared to the total trajectory time of 100 days, we can see that even with short time horizons the MPC trajectories are capable of making it to the goal location

1 Referee 1

The manuscript is concerned with using model predictive control to obtain energy-efficient trajectories for vehicles which are subject to spatially and temporally varying background flows (wind, ocean currents, etc). Key aspects of the manuscript are the use of very short time horizons and the observation for the test case that the resulting efficient trajectories are meaningfully exploiting structures in the background flow.

Overall the manuscript is clear and easy to follow. However, the conclusions are weakened somewhat by the consideration of a single 2D test case where it is assumed that the background flow field is known perfectly and that the vehicle does not meaningfully influence field. Whilst these set of assumptions are reasonable for ocean vehicles, they are likely to be deficient for any aerial applications (several of which are mentioned in the abstract, introduction, and conclusion).

The manuscript would therefore benefit greatly from a 3D test case (which should be a straightforward extension) with a more erratic background field. Or, alternatively, an attempt to incorporate measurement noise into the background field. Having one, or ideally both of these, would go a long way towards enhancing the impact of the manuscript.

Author response: Thank you for your kind words and positive assessment of our review. We are especially glad that you found it to be clear and easy to follow. We are also grateful for your comments and suggestions below, and we have made every attempt to address these all in the revision, with detailed responses below.

We have taken your suggestion and considered both a new 3D test case (the chaotic ABC flow) and also a real-world 2D ocean flow in the Gulf of Mexico, which has multiscale, turbulent dynamics. The main results of the paper generalize in these more complex flows, and we have reported these findings in the new Section 5. We hope that this addresses your main concern, and we believe this has considerably strengthened the revised manuscript.

1. p5 Eq (2.3) is awkward to unpick and would benefit from introducing a grid-spacing variable h_{ij} .

Author response: This is a good point. We have modified this equation to be more clear, as follows:

$$\begin{aligned} \left(\mathbf{D}\Phi_{t_0}^{t_0+T} \right)_{i,j} &= \begin{bmatrix} \frac{\Delta x_i(t_0+T)}{\Delta x_i(t_0)} & \frac{\Delta x_j(t_0+T)}{\Delta y_j(t_0)} \\ \frac{\Delta y_i(t_0+T)}{\Delta x_i(t_0)} & \frac{\Delta y_j(t_0+T)}{\Delta y_j(t_0)} \end{bmatrix} \\ &= \begin{bmatrix} \frac{x_{i+1,j}(t_0+T) - x_{i-1,j}(t_0+T)}{x_{i+1,j}(t_0) - x_{i-1,j}(t_0)} & \frac{x_{i,j+1}(t_0+T) - x_{i,j-1}(t_0+T)}{y_{i,j+1}(t_0) - y_{i,j-1}(t_0)} \\ \frac{y_{i+1,j}(t_0+T) - y_{i-1,j}(t_0+T)}{x_{i+1,j}(t_0) - x_{i-1,j}(t_0)} & \frac{y_{i,j+1}(t_0+T) - y_{i,j-1}(t_0+T)}{y_{i,j+1}(t_0) - y_{i,j-1}(t_0)} \end{bmatrix}, \end{aligned} \tag{2.3}$$

2. p6 Wording: ‘More broadly, FTLE has been used to coherent structures’

Author response: Thanks for catching this. We have fixed this in the revised manuscript.

2 Referee 2

Inspired by the application of mobile sensors, the paper centers around the application of finite horizon model predictive control (MPC) and finite time Lyapunov exponents (FTLE) using simple test problems and synthetic flow fields. The key premise of the paper is to find mobile sensor trajectories for a relatively short horizon using limited or partial knowledge of the background flow field. The authors attempt to understand the sensitivity of the trajectories and estimate an optimal range where energy-efficient trajectories are found. Overall the paper is well written and has some scientific relevance on unsteady flow prediction & control of mobile sensors. My main criticism is with regard to the novelty and practical relevance of the presented results. Some specific concerns are as follows:

Author response: We appreciate that you found the paper well written and to have relevance. We also very much appreciate the suggestions below to improve the practical value with more sophisticated examples and an improved discussion. We have made efforts to address these issues below, and we believe we have strengthened the revised manuscript considerably.

1. I'm not clear about the novelty and archival value of this paper. I want the authors to articulate their specific contributions and their practical demonstration. MPC and FTLE are standard approaches in control theory and dynamical systems. The codes and libraries are easily available. The results are based on toy problems and they lack generality to real world situation.

Author response: We appreciate the referee's comment, and realize that this can be strengthened and clarified in the revised manuscript. We do agree that the original results were based on toy problems, which limited the applicability and generality of the results. In the revised version, we have added test cases with two new flow fields that are considerably more complex to demonstrate the universality of our findings. In addition to these entirely new, more sophisticated examples, we outline the novelty of the core results here. The novelty and archival value of this paper reside in the following contributions. Firstly, this carefully and thoroughly explores the relationship between energy efficient trajectories found by finite-horizon model predictive control and FTLE structures. Existing works on this subject have focused on either the globally energy-optimal trajectories or optimization with a receding horizon. These methods require knowledge of the flow field at all times, which is not feasible in many practical applications. Finite-horizon MPC, on the other hand, only requires knowledge about flow for the prediction horizon. Secondly, We provided a comprehensive analysis of the resulting MPC trajectories under different optimization parameter settings, leading to several new findings, including bifurcations of the trajectory into periodic orbits around the goal location with periods that are consistent with the periodicity of the flow field; the break-off from the Pareto trade-off curves as the agent's motions become dominated by the flow field; and the agent's tendency to move perpendicular to the flow direction when the actuation penalty is small.

Although much of the analysis were based on an analytic double-gyre flow, we chose flow and optimization parameters that are representative of several different real-world situations and mobile sensor types. For instance, smaller R/Q values correspond to more energetic platforms such as autonomous underwater vehicles while larger R/Q values correspond to less-actuated alternatives such as the ocean gliders or floats. However, we do agree that these results are limited in generality, and we believe that the two new test fields address this concern and strengthen the manuscript.

2. In Section 3, why the flow field is considered simple, instead of real flow field from CFD or experimental data? The current results have no relevance to any practical problem. Unfortunately, the presented results lack physical interpretation. They are several trajectories presented but it's not clear why and how? Also there are no verification and validation of the claims the authors have made about the prediction horizon and the energy optimal trajectories.

Author response: We appreciate the referee challenging us to consider more sophisticated flows that are more relevant to practical situations. We have added two new more challenging test flows: a 3D chaotic flow field (the ABC flow), and data from the Gulf of Mexico. These results are presented in the new Section 5, and corroborate many of the findings with the double gyre.

However, examining a simple flow thoroughly does still have value to build intuition. We chose the double-gyre flow for our investigation because it well captures the chaotic mixing that is often observed in real-world oceanic and atmospheric applications. The double-gyre flow also has an analytic expression, making it ideal for us to focus on the dependency of energy-optimal trajectories on the unsteady transport behaviors. Further, this example is well studied as a benchmark problem for FTLE.

We see the referee's point that "energy optimal" is not the best terminology, and have opted for "energy efficient" in several key places throughout the manuscript.

We agree that a truly turbulent flow field better capture real-world scenarios, and so we have considered the two more sophisticated flows mentioned above. We believe that these examples have considerably strengthened the results in the paper.

3. While the ocean flow is turbulent, the authors should discuss the impact of turbulence into their prediction horizons and the energy efficiency.

Author response: This is a very good point. We have added results from the unsteady 3D ABC flow as well as from turbulent ocean data in the Gulf of Mexico. In addition to these new results, we also discuss the impact of turbulence on the prediction horizon and energy efficiency as suggested.

It will be important to characterize how multiscale turbulence will affect the prediction horizon, as uncertainties will be magnified making it challenging to forecast flow structures. These multiscale structures will also impact energy efficiency, both through the forecast uncertainty, but also through making optimal paths more circuitous.

3 Referee 3

The topic of the manuscript is important, the text is well written, and the figures are of good quality. Linking energy-optimal paths (and time-optimal paths) with Lagrangian Coherent Structures (LCS) and FTLE fields is interesting, and should be studied both theoretically and in actual applications. As the LCS correspond to the more robust and often stronger flow structures, it is normal that energy-optimal paths and time-optimal paths depend on these flow structures. In fact, it is likely that for vehicles with variable speed, all optimal paths that are affected by the unsteady flow environment (energy-optimal, time-optimal, or any other quantity related to velocity) will depend on the more robust and dominant flow structures. Saying that FTLE is linked to optimality is basically saying that optimal paths in a strong-enough dynamic flow will be affected by the dominant structures of the flow. What would be much more interesting is to determine and explain what are the properties of the LCSs that directly relate to specific optimality properties, for example what LCS property relates to energy-optimality? time-optimality? or both? The authors attempt to address this question for energy-optimality but more quantitative results, more complex examples, and more detailed analyses are needed. We recommend to revise the manuscript.

Author response: We are grateful for the referee's positive comments, and are especially encouraged that you find it to be important and well written. We also appreciate the constructive comments here and below, and we have made several efforts to address these throughout, which we believe has considerably improved the revised manuscript.

1. Some readers may say that the authors claim more at the start of the manuscript than they actually achieve as results. The authors could likely tone down the abstract and introduction claims or add more results in the following sections.

Author response: This is a good point, and we appreciate the frank assessment. We have added two new test flows that are considerably more sophisticated to strengthen our claims: a 3D chaotic ABC flow and 2D turbulent ocean data from the Gulf of Mexico. These are analyzed in the new Section 5, and the results indicate that our findings generalize to other flows.

2. Only a single and relatively simple example is provided. The authors should likely consider other flow fields and other start times and positions.

Author response: We agree, and we have added two new flows, as discussed above.

3. The authors should update their title and abstract and introduction since the cost function also penalizes the instantaneous and final distances with respect to the goal, and also because the MPC solution is not guaranteed to provide the exact energy-optimal control up to the arrival time. Using energy-efficient throughout the paper would be more adequate. Updates should be made in many sentences throughout the paper. It is only for a time-horizon that is equal to the arrival time that the MPC solution would minimize the energy cost. Also, as set-up, the solution is not guaranteed to reach the target endpoint.

Author response: We appreciate the referee’s point here, and we agree that energy-efficient would be more appropriate in several cases than energy-optimal. We have changed this in the title and several places throughout the manuscript.

4. The links between the FTLE fields and the optimal paths are weak. The authors should determine and explain quantitatively what are the properties of the LCSs that directly relate to specific optimality properties, for example what LCS property relates to energy-optimality? Or time-optimality? It would then be interesting to highlight some of the deeper mathematical/ physical reasons in the examples, so as to explain why some of the observed phenomena take place. For example, the limit for a vehicle speed going to zero could be useful.

Author response: Thank you for this suggestion, we too found it important to provide a discussion on this matter. We have attempted to better elucidate the link between FTLE and optimal trajectories in the discussions in the main text.

We also found that locations where the sensor spends most energy correlated with the presence of repelling hyperbolic LCS. These findings could be better understood by recalling that repelling LCS are defined as material barriers in the flow field. This explains the expenditure of more energy close to the repelling LCS, as energy needs to be spent to overcome barrier for movement. Furthermore, as the energy spent for movement goes to zero, due to cost penalization, the sensor behaves similar to a passive tracer. It is known that the short term dynamics of passive tracers are governed by movement with the FTLE. This suggests that these connections hold for energy optimality.

Adding to the discussion in the text, the task of ocean navigation is generally not time sensitive, we have not investigated explicit penalization on time spent away from the goal. Moreover, given that we are explicitly penalizing energy in our cost function, our observed results (and investigations by past papers) point strongly in the direction of energy optimality.

5. The details and variables of the MPC procedures should be clarified. For example, for the variables, the arrival time (not defined), the finite-time horizon T_H , the planning time or frequency, the receding time, energy measure, relative size of $Q_1 T_H$ and Q_2 , etc. should be discussed. Similarly, the computational properties, e.g. spatial and temporal discretization, the optimization scheme, etc, should be discussed, e.g. in or after section 3(c). The computational costs and accuracy should also be mentioned, how efficient is the MPC approach compared to other methods? All of the above should then be mentioned in the figures. Some other specific examples and questions:
 - a. The results section 4(d) should be summarized or introduced sooner, this would warn the reader that some of the questions are addressed later.
 - b. Time horizon for FTLE fields. Is it always the same as the time horizon for MPC?
 - c. Re-planning frequency for MPC: The kinks in fig 2 seem to point to the re-planning frequency being too high. Are some of the results (e.g., the extra loop R/Q = 2) just a

result of a poor choice of the re-planning frequency? If this was high enough, shouldn't the vehicle be able to better time its exit?

d. In figure 3, the authors may want to clarify their results and the rationale for choosing T_H .

Author response: We thank the referee for pointing out these missing details, since it is important to us to provide all the details to ensure our results are reproducible. In the revised Section 3(c), we have added the choice of replanning frequency (added an explanation in the introduction on MPC) and total simulation steps used. We also mentioned the optimization scheme used in our implementation. Depending on the context, the figures use different time horizons, therefore for each figure, the T_H has been mentioned.

(a) This is a good suggestion. We have added a sentence pointing the reader to Section 4(d) for the results of the thorough parameter sweep.

The MPC results for this thorough parameter sweep are presented in Section 4(d) and summarized in Figure 7.

(b) The time horizon for FTLE is fixed at 15 units throughout the paper, while the MPC time horizon is varied to see the effect of changing the horizon (e.g., Figure 3, Figure 7).

(c) This is an interesting question, and is something we have wondered about ourselves. We have used different sampling and replanning frequencies, and these results appear to be robust to this. This is what led us to investigate several nearby time horizons in Figure 3 and compare the R/Q in Figure A.1. In these figures, we see that the right loop observed in Figure 2 is actually due to a sensitivity in the trajectory with respect to time horizon, where the trajectory gets trapped in the right gyre for one period, before planning a path to cross into the left gyre.

(d) This plot is closely related to the question (c) above, and was included to investigate the sensitivity of the trajectory with T_H . We have added a clarification in the caption:

By varying the time horizon, T_H , we see that the extra loop in Figure 2 is due to a sensitivity of the planned path with respect to T_H , where the agent becomes stuck in the right gyre for lower T_H .

6. For the given double gyre case of interest, the start point for all optimal paths was kept fixed. It would be interesting to see how the connections highlighted between the MPC paths and FTLE would vary with different start (and end) points.

Author response: We have now included Figure 9 with different start points, and a separate Figure 10 for different end points. We observe that changing the start points does not have much impact on the final orbit the sensor settles on. This shows that the choice of initial condition does not greatly change how the LCS are used from the initial cases we studied. In Figure 10, we see that changing the goal location *does* change the final orbit, but sensors on these orbits still do move with the LCS. There

Figure 9: MPC trajectories with varying R/Q ratio, starting from different initial conditions with the same goal location. We find that eventually, the trajectories converge to similar periodic orbits thereby using the LCS in similar ways to orbit around the goal location despite having different transients.

are some regions of space that are much more difficult to loiter around because of the unsteady background flow. We believe that this important point is also more clear in the revised manuscript. The new Figures 9 and 10 are reproduced here.

Different Start and End Locations

Figure 9 shows the MPC optimized trajectories for six different starting locations along the right and lower boundaries of the domain. Although the paths have different initial transients, the trajectories evolve onto the same periodic loitering orbits around the goal state, indicating that they are ultimately leveraging similar flow structures. Similarly, Figure 10 shows the MPC optimized trajectories for several different goal states. Some goal states are much more difficult to reach than others, because of the strong unsteady background flow field. With more aggressive control ($R/Q = 1$) the trajectories generally form tighter loitering orbits. However, for the case when the goal state is in the upper middle of the domain, it is clear that none of the MPC trajectories are able to find a suitable loitering pattern. This diversity of orbits highlights the importance of choosing a suitable goal location, which would likely involve a higher level of planning.

7. Section 4(a): a. “Correspond to bifurcations in the orbit” - Could the authors elaborate and explain these bifurcations in the Figure captions? b. Results in section 4(a) seem to be more about the behavior of MPC with varying parameters on this specific dynamical system rather than its relation to the LCS. The observation at the end of the section for result 4(a) about the point hovering at the intersection of the two FTLE ridges is indeed interesting and could be elaborated further. Is there a physical reason why it should be at the intersection? Why do we see this behavior only for certain parameters? Would we see this behaviour even when the destination is not at the center of a gyre? c. Section 4(a): The observation about the extra right loop for

Figure 10: MPC trajectories to different goal locations with varying R/Q ratio for the same initial condition. We find that in the case of placing the goal near the middle to bottom region (top left and top middle plots) of double gyre, the sensor is able to form stable orbits near the attracting LCS, where the sensor moves with the base of the attracting LCS. However, placing the goal near the repelling LCS causes more difficulty for MPC in forming small stable orbits (top right). The bottom plots show that it is possible to form small periodic orbits in the corners of the double gyre flow field when the goal is placed close to them.

$R/Q = 2$ seems more like a product of T_H and the re-planning frequency. Could the authors study this?

Author response: Thank you for these comments and for carefully reading the paper. (a) We have made the changes in the caption of Figure 2 to clarify this point. Specifically we added “An example of this is when R/Q is changed from 25 to 26, we observe a major change in the shape of the final orbit around the goal, as opposed to the minor change from $R/Q = 15$ to 25, where the final eye-shaped orbit only gradually increases in size. ”

(b) We observe this behaviour for orbits formed with low R/Q near saddle corner fixed points in the double gyre. This behaviour thus seems to occur closer to the fixed points that exist in a steady double gyre.

(c) This is a good observation, and this is something we did investigate in Figure 3, where we show that the extra loop is related to the time horizon.

8. Section 4(b): a. This observation is indeed relevant to the paper and interesting. Would it be possible to also show the behavior when crossing an attracting ridge?

Author response: This is a good question. As far as our observations in the double gyre case, the correlation between energy spent and movement across ridge holds only

in the case of hyperbolic repelling FTLE ridges. We have added a line in section 4(b) mentioning this.

9. Section 4(c): a. The links to FTLEs could be strengthened. The behavior intuitively makes sense based on how MPC works. It is not clear that FTLEs and energy-optimal are more connected than FTLE and time-optimal. b. The tendency to form orbits makes sense given that the destination is in the center of a gyre. One potential thing to explore would be: What is the threshold to form orbits based on the location of the destination? One interesting observation could be the system likes forming orbits when surrounded by an FTLE ridge (like in the center of a gyre) but would not form orbits when the destination sits on a repelling ridge

Author response: We were able to observe your last point when the goal was placed close to a repelling fixed point. Close to such a point, the sensor hops around and finally forms an orbit around the centre of a gyre or keeps shooting towards the repelling point. However, close to an attracting fixed point, the sensor moves with the attracting LCS. The results of this study were summarized in Figure 10. We also observe that stable orbits exist at the corner saddle fixed points of the double gyre.

10. The results in Figure 4 are quite compelling for the effects of FTLE on optimal paths (energy or time optimal for that matter), for vehicles with variable speeds. It might be interesting to see this phenomenon for other paths (or different points along the path) where an FTLE ridge is crossed to further highlight the proposed correlation. Perhaps (inspired by Figure 8) a study can be completed where an ensemble of different start points are considered and the distribution of the control effort and cost-to-go jumps are considered at all points when an FTLE ridge is crossed. In summary, it would be good to see how the results from Figure 4 hold for more than just this crossing of a FTLE ridge shown (Section 4(b)).

Author response: We are glad that you find this compelling – we were also encouraged by this result. In the double gyre flow field, we observe the rise in energy only at the hyperbolic FTLE ridges in the middle of the gyre which doesn't seem to occur at the shear-dominated FTLE ridges. All the trajectories in the paper, which are color coded, tend to be yellow (high energy) in the region around the middle of the double gyre flow field where this repelling LCS exists. This was also observed in the new Gulf of Mexico test case as summarized in Figure 13 and 14. We hope that this further strengthens this connection.

11. While the results in Section 4c are interesting, the connection to how fluid coherent structures cause this motion should be addressed and discussed in detail (since this is the focus of the paper as highlighted in the introduction). In particular, the observations in this section of the results seem to be highly particular to the double gyre case considered, and it is unclear how this would generalize to other flows.

Author response: We agree that these periodic orbits are interesting. Although it does seem intuitive that these may be particular to the double gyre case, they are also strongly observed in the new 3D ABC flow, and to a lesser extent in the Gulf of Mexico data set. We believe this provides evidence that these may be more generally found. We also agree that this would be interesting to explore in more detail, and we have added a short statement to this effect at the end of this section.

In the other example flows observed below, similar periodic orbits are observed, where the agent *loiters* around the goal state. It will be interesting to investigate these orbits in more detail, including the classes of flows they exist in, and the conditions under which they bifurcate.

12. What is the time horizon chosen for the LCS plots? Is it always the same as the MPC T_H ?

Author response: The time horizon for FTLE throughout the paper is kept to be 15 units, while the MPC time horizon is being changed depending on the context of the plot or text.

13. Section 1: “An predecessor of this was the planning of space missions” should most likely be “A predecessor ...”

Author response: Good catch. We have fixed this.

14. Figure 1 caption: “and colored-coded based on ...” should most likely be “and color-coded ...”

Author response: Thank you. We have fixed this.

15. Section 2(b): “The controller enacts this optimal actuation policy for a short time, often for a single time step, and then the optimization problem is recomputed AND initialized at the current state.”

Author response: Corrected.

16. 3(c) State tracking error: - “ $x - x_{goal}$ ” instead of “ $x_{start} - x_{goal}$ ”

Author response: Corrected.

17. Figure 4 caption: “Here, unlike in Figure 2, The summations ...” should most likely be “Here, unlike in Figure 2, the summations ...”

Author response: Corrected

18. Figure 4 caption: “There is a correlation between spike in both ...” should most likely be “There is a correlation between the spike in both ...”

Author response: Corrected.

References

- [1] Bellingham JG, Rajan K. 2007 Robotics in remote and hostile environments. *Science* **318**, 1098–1102.
- [2] Wynn RB, Huvenne VA, Le Bas TP, Murton BJ, Connelly DP, Bett BJ, Ruhl HA, Morris KJ, Peakall J, Parsons DR, Sumner EJ, Darby SE, Dorrell RM, Hunt JE. 2014 Autonomous Underwater Vehicles (AUVs): Their past, present and future contributions to the advancement of marine geoscience. *Marine Geology* **352**, 451–468.
- [3] Rhoads B, Mezić I, Poje AC. 2013 Minimum time heading control of underpowered vehicles in time-varying ocean currents. *Ocean Engineering* **66**, 12–31.
- [4] Fossum TO, Fragoso GM, Davies EJ, Ullgren JE, Mendes R, Johnsen G, Ellingsen I, Eidsvik J, Ludvigsen M, Rajan K. 2019 Toward adaptive robotic sampling of phytoplankton in the coastal ocean. *Science Robotics* **4**.
- [5] Chai F, Johnson KS, Claustre H, Xing X, Wang Y, Boss E, Riser S, Fennel K, Schofield O, Sutton A. 2020 Monitoring ocean biogeochemistry with autonomous platforms. *Nature Reviews Earth & Environment* **1**, 315–326.
- [6] Zhang Y, Ryan JP, Hobson BW, Kieft B, Romano A, Barone B, Preston CM, Roman B, Raanan BY, Pargett D et al.. 2021 A system of coordinated autonomous robots for Lagrangian studies of microbes in the oceanic deep chlorophyll maximum. *Science Robotics* **6**.
- [7] Kularatne D, Bhattacharya S, Hsieh MA. 2016 Time and Energy Optimal Path Planning in General Flows.. In *Robotics: Science and Systems*.
- [8] Rao D, Williams SB. 2009 Large-scale path planning for underwater gliders in ocean currents. In *Australasian conference on robotics and automation (ACRA)* pp. 2–4.
- [9] Subramani DN, Lermusiaux PF. 2016 Energy-optimal path planning by stochastic dynamically orthogonal level-set optimization. *Ocean Modelling* **100**, 57–77.
- [10] Yilmaz NK, Evangelinos C, Lermusiaux PF, Patrikalakis NM. 2008 Path planning of autonomous underwater vehicles for adaptive sampling using mixed integer linear programming. *IEEE Journal of Oceanic Engineering* **33**, 522–537.
- [11] Lermusiaux PF. 2007 Adaptive modeling, adaptive data assimilation and adaptive sampling. *Physica D: Nonlinear Phenomena* **230**, 172–196.
- [12] Bhatta P, Fiorelli E, Lekien F, Leonard NE, Paley D, Zhang F, Bachmayer R, Davis RE, Fratantoni DM, Sepulchre R. 2005 Coordination of an underwater glider fleet for adaptive ocean sampling. In *Proc. International Workshop on Underwater Robotics, Int. Advanced Robotics Programmed (IARP), Genoa, Italy*.

- [13] Leonard NE, Paley DA, Lekien F, Sepulchre R, Fratantoni DM, Davis RE. 2007 Collective motion, sensor networks, and ocean sampling. *Proceedings of the IEEE* **95**, 48–74.
- [14] Fiorelli E, Leonard NE, Bhatta P, Paley DA, Bachmayer R, Fratantoni DM. 2006 Multi-AUV control and adaptive sampling in Monterey Bay. *IEEE Journal of Oceanic Engineering* **31**, 935–948.
- [15] Leonard NE, Graver JG. 2001 Model-based feedback control of autonomous underwater gliders. *IEEE Journal of Oceanic Engineering* **26**, 633–645.
- [16] Lipinski D, Mohseni K. 2010 Cooperative control of a team of unmanned vehicles using smoothed particle hydrodynamics. In *AIAA Guidance, Navigation, and Control Conference* p. 8316.
- [17] Lipinski D, Mohseni K. 2014 Feasible area coverage of a hurricane using micro-aerial vehicles. In *AIAA Atmospheric Flight Mechanics Conference* p. 0894.
- [18] Lipinski D, Mohseni K. 2011 A master-slave fluid cooperative control algorithm for optimal trajectory planning. In *2011 IEEE International Conference on Robotics and Automation* pp. 3347–3351. IEEE.
- [19] Song Z, Lipinski D, Mohseni K. 2017 Multi-vehicle cooperation and nearly fuel-optimal flock guidance in strong background flows. *Ocean Engineering* **141**, 388–404.
- [20] Song Z, Mohseni K. 2015 Anisotropic active Lagrangian particle swarm control in a meandering jet. In *2015 54th IEEE Conference on Decision and Control (CDC)* pp. 240–245. IEEE.
- [21] Gunnarson P, Mandralis I, Novati G, Koumoutsakos P, Dabiri JO. 2021 Learning Efficient Navigation in Vortical Flow Fields. *arXiv preprint arXiv:2102.10536*.
- [22] Biferale L, Bonaccorso F, Buzzicotti M, Clark Di Leoni P, Gustavsson K. 2019 Zermelo’s problem: Optimal point-to-point navigation in 2D turbulent flows using reinforcement learning. *Chaos: An Interdisciplinary Journal of Nonlinear Science* **29**, 103138.
- [23] Buzzicotti M, Biferale L, Bonaccorso F, di Leoni PC, Gustavsson K. 2021 Optimal control of point-to-point navigation in turbulent time-dependent flows using reinforcement learning. .
- [24] Inanc T, Shadden SC, Marsden JE. 2005 Optimal trajectory generation in ocean flows. In *Proceedings of the 2005, American Control Conference, 2005.* pp. 674–679. IEEE.
- [25] Zhang W, Inanc T, Ober-Blobaum S, Marsden JE. 2008 Optimal trajectory generation for a glider in time-varying 2D ocean flows B-spline model. In *2008 IEEE International Conference on Robotics and Automation* pp. 1083–1088. IEEE.
- [26] Senatore C, Ross SD. 2008 Fuel-efficient navigation in complex flows. In *2008 American Control Conference* pp. 1244–1248. IEEE.

- [27] Heckman CR, Hsieh MA, Schwartz IB. 2016 Controlling basin breakout for robots operating in uncertain flow environments. In *Experimental Robotics* pp. 561–576. Springer.
- [28] Haller G. 2002 Lagrangian coherent structures from approximate velocity data. *Physics of fluids* **14**, 1851–1861.
- [29] Haller G. 2005 An objective definition of a vortex. *Journal of Fluid Mechanics* **525**, 1–26.
- [30] Shadden SC, Lekien F, Marsden JE. 2005 Definition and properties of Lagrangian coherent structures from finite-time Lyapunov exponents in two-dimensional aperiodic flows. *Physica D: Nonlinear Phenomena* **212**, 271–304.
- [31] Shadden SC, Lekien F, Paduan JD, Chavez FP, Marsden JE. 2009 The correlation between surface drifters and coherent structures based on high-frequency radar data in Monterey Bay. *Deep Sea Research Part II: Topical Studies in Oceanography* **56**, 161–172.
- [32] Shadden SC. 2011 Lagrangian coherent structures. *Transport and Mixing in Laminar Flows: From Microfluidics to Oceanic Currents* pp. 59–89.
- [33] Haller G. 2015 Lagrangian coherent structures. *Annual Review of Fluid Mechanics* **47**, 137–162.
- [34] Sudharsan M, Brunton SL, Riley JJ. 2016 Lagrangian coherent structures and inertial particle dynamics. *Physical Review E* **93**, 033108.
- [35] Brunton SL, Rowley CW. 2010 Fast computation of finite-time Lyapunov exponent fields for unsteady flows. *Chaos: An Interdisciplinary Journal of Nonlinear Science* **20**, 017503.
- [36] Lipinski D, Mohseni K. 2010 A ridge tracking algorithm and error estimate for efficient computation of Lagrangian coherent structures. *Chaos* **20**, 017503.
- [37] Haller G. 2011 A variational theory of hyperbolic Lagrangian coherent structures. *Physica D: Nonlinear Phenomena* **240**, 574–598.
- [38] Senatore C, Ross SD. 2011 Detection and characterization of transport barriers in complex flows via ridge extraction of the finite time Lyapunov exponent field. *International Journal for Numerical Methods in Engineering* **86**, 1163–1174.
- [39] Farazmand M, Haller G. 2012 Computing Lagrangian coherent structures from their variational theory. *Chaos: An Interdisciplinary Journal of Nonlinear Science* **22**, 013128.
- [40] BozorgMagham AE, Ross SD, Schmale III DG. 2013 Real-time prediction of atmospheric Lagrangian coherent structures based on forecast data: An application and error analysis. *Physica D: Nonlinear Phenomena* **258**, 47–60.

- [41] Tallapragada P, Ross SD. 2013 A set oriented definition of finite-time Lyapunov exponents and coherent sets. *Communications in Nonlinear Science and Numerical Simulation* **18**, 1106–1126.
- [42] Serra M, Haller G. 2016 Objective Eulerian coherent structures. *Chaos: An Interdisciplinary Journal of Nonlinear Science* **26**, 053110.
- [43] Wilson MM, Peng J, Dabiri JO, Eldredge JD. 2009 Lagrangian coherent structures in low Reynolds number swimming. *Journal of Physics: Condensed Matter* **21**, 204105.
- [44] Shadden SC, Taylor CA. 2008 Characterization of coherent structures in the cardiovascular system. *Annals of Biomedical Engineering* **36**, 1152–1162.
- [45] Forgoston E, Bianco S, Shaw LB, Schwartz IB. 2011 Maximal sensitive dependence and the optimal path to epidemic extinction. *Bulletin of mathematical biology* **73**, 495–514.
- [46] Tallapragada P, Ross SD, Schmale III DG. 2011 Lagrangian coherent structures are associated with fluctuations in airborne microbial populations. *Chaos: An Interdisciplinary Journal of Nonlinear Science* **21**, 033122.
- [47] Rockwood MP, Loiselle T, Green MA. 2019 Practical concerns of implementing a finite-time Lyapunov exponent analysis with under-resolved data. *Experiments in Fluids* **60**, 74.
- [48] Rockwood MP, Green MA. 2019 Real-time identification of vortex shedding in the wake of a circular cylinder. *AIAA Journal* **57**, 223–238.
- [49] Koon WS, Lo MW, Marsden JE, Ross SD. 2006 Dynamical systems, the three-body problem and space mission design. *California Institute of Technology, Pasadena, CA, USA*.
- [50] Lagor FD, Paley DA. 2014 Active Singularities for Multivehicle Motion Planning in an N-Vortex System. In *International Conference on Dynamic Data-Driven Environmental Systems Science* pp. 334–346. Springer.
- [51] Lagor FD, Ide K, Paley DA. 2015 Touring invariant-set boundaries of a two-vortex system using streamline control. In *2015 54th IEEE Conference on Decision and Control (CDC)* pp. 2217–2222. IEEE.
- [52] Ramos A, García-Garrido V, Mancho A, Wiggins S, Coca J, Glenn S, Schofield O, Kohut J, Aragon D, Kerfoot J et al.. 2018 Lagrangian coherent structure assisted path planning for transoceanic autonomous underwater vehicle missions. *Scientific Reports* **8**, 1–9.
- [53] Holmes P, Guckenheimer J. 1983 *Nonlinear oscillations, dynamical systems, and bifurcations of vector fields* vol. 42 *Applied Mathematical Sciences*. Berlin, Heidelberg: Springer-Verlag.

- [54] Dellnitz M, Froyland G, Junge O. 2001 The algorithms behind GAIO—Set oriented numerical methods for dynamical systems. In *Ergodic theory, analysis, and efficient simulation of dynamical systems* pp. 145–174. Springer.
- [55] Froyland G. 2005 Statistically optimal almost-invariant sets. *Physica D: Nonlinear Phenomena* **200**, 205–219.
- [56] Froyland G, Padberg K. 2009 Almost-invariant sets and invariant manifolds – Connecting probabilistic and geometric descriptions of coherent structures in flows. *Physica D* **238**, 1507–1523.
- [57] Froyland G, Santitissadeekorn N, Monahan A. 2010 Transport in time-dependent dynamical systems: Finite-time coherent sets. *Chaos* **20**, 043116–1–043116–16.
- [58] Kelley DH, Allshouse MR, Ouellette NT. 2013 Lagrangian coherent structures separate dynamically distinct regions in fluid flows. *Physical Review E* **88**, 013017.
- [59] Olascoaga MJ, Rypina I, Brown MG, Beron-Vera FJ, Koçak H, Brand LE, Halliwell G, Shay LK. 2006 Persistent transport barrier on the West Florida Shelf. *Geophysical research letters* **33**.
- [60] Beron-Vera FJ, Olascoaga MJ, Goni G. 2008 Oceanic mesoscale eddies as revealed by Lagrangian coherent structures. *Geophysical Research Letters* **35**.
- [61] Beron-Vera FJ, Olascoaga MJ, Haller G, Farazmand M, Triñanes J, Wang Y. 2015 Dissipative inertial transport patterns near coherent Lagrangian eddies in the ocean. *Chaos: An Interdisciplinary Journal of Nonlinear Science* **25**, 087412.
- [62] Lekien F, Coulliette C, Mariano AJ, Ryan EH, Shay LK, Haller G, Marsden JE. 2005 Pollution release tied to invariant manifolds: A case study for the coast of Florida. *Physica D* **210**, 1–20.
- [63] Green MA, Rowley CW, Haller G. 2007 Detection of Lagrangian coherent structures in 3D turbulence.. *Journal of Fluid Mechanics* **572**, 111–120.
- [64] Franco E, Pekarek DN, Peng J, Dabiri JO. 2007 Geometry of unsteady fluid transport during fluid-structure interactions. *Journal of Fluid Mechanics* **589**, 125–145.
- [65] Padberg K, Hauff T, Jenko F, Junge O. 2007 Lagrangian structures and transport in turbulent magnetized plasmas. *New Journal of Physics* **9**, 400.
- [66] Mathur M, Haller G, Peacock T, Ruppert-Felsot JE, Swinney HL. 2007 Uncovering the Lagrangian skeleton of turbulence. *Physical Review Letters* **98**, 144502–1–144502–4.
- [67] Peng J, Dabiri JO. 2008 The ‘upstream wake’ of swimming and flying animals and its correlation with propulsive efficiency. *The Journal of Experimental Biology* **211**, 2669–2677.

- [68] Rockwood MP, Taira K, Green MA. 2016 Detecting Vortex Formation and Shedding in Cylinder Wakes Using Lagrangian Coherent Structures. *AIAA Journal* **55**, 15–23.
- [69] Garcia CE, Prett DM, Morari M. 1989 Model predictive control: Theory and practice—A survey. *Automatica* **25**, 335–348.
- [70] Mayne DQ, Rawlings JB, Rao CV, Sokaert PO. 2000 Constrained model predictive control: Stability and optimality. *Automatica* **36**, 789–814.
- [71] Camacho EF, Alba CB. 2013 *Model Predictive Control*. Springer Science & Business Media.
- [72] Kaiser E, Kutz JN, Brunton SL. 2018 Sparse identification of nonlinear dynamics for model predictive control in the low-data limit. *Proceedings of the Royal Society A* **474**, 20180335.
- [73] Hovgard TG, Larsen LF, Jørgensen JB, Boyd S. 2012 Fast nonconvex model predictive control for commercial refrigeration. *IFAC Proceedings Volumes* **45**, 514–521.
- [74] Jiang S, Jin Ff, Ghil M. 1995 Multiple equilibria, periodic, and aperiodic solutions in a wind-driven, double-gyre, shallow-water model. *Journal of Physical Oceanography* **25**, 764–786.
- [75] Speich S, Ghil M. 1994 Interannual variability of the mid-latitude oceans: A new source of climate variability. *Sistema Terra* **3**, 459.
- [76] Speich S, Dijkstra H, Ghil M. 1995 Successive bifurcations in a shallow-water model applied to the wind-driven ocean circulation. *Nonlinear Processes in Geophysics* **2**, 241–268.
- [77] Andersson JAE, Gillis J, Horn G, Rawlings JB, Diehl M. 2019 CasADi – A software framework for nonlinear optimization and optimal control. *Mathematical Programming Computation* **11**, 1–36.
- [78] Risbeck MJ, Rawlings JB. 2015 MPCTools: Nonlinear model predictive control tools for CasADi (Python interface). <https://bitbucket.org/rawlings-group/mpc-tools-casadi>.
- [79] Budišić M, Siegmund S, Thai Son D, Mezic I. 2016 Mesochronic classification of trajectories in incompressible 3D vector fields over finite times. .
- [80] Ricca R. 2001 *An Introduction to the Geometry and Topology of Fluid Flows*.
- [81] Michini M, Mallory K, Larkin D, Hsieh MA, Forgoston E, Yecko PA. 2013 An experimental testbed for multi-robot tracking of manifolds and coherent structures in flows. In *Dynamic Systems and Control Conference* vol. 56130 p. V002T32A002. American Society of Mechanical Engineers.

- [82] Michini M, Hsieh MA, Forgoston E, Schwartz IB. 2014 Robotic tracking of coherent structures in flows. *IEEE Transactions on Robotics* **30**, 593–603.
- [83] Mallory K, Hsieh MA, Forgoston E, Schwartz IB. 2013 Distributed allocation of mobile sensing swarms in gyre flows. *Nonlinear Processes in Geophysics* **20**, 657–668.
- [84] BozorgMagham AE, Ross SD. 2015 Atmospheric Lagrangian coherent structures considering unresolved turbulence and forecast uncertainty. *Communications in Nonlinear Science and Numerical Simulation* **22**, 964–979.
- [85] Balasuriya S. 2020 Uncertainty in finite-time Lyapunov exponent computations. *Journal of Computational Dynamics* **7**, 313.
- [86] Luchtenburg DM, Brunton SL, Rowley CW. 2014 Long-time uncertainty propagation using generalized polynomial chaos and flow map composition. *Journal of Computational Physics* **274**, 783–802.
- [87] Taira K, Brunton SL, Dawson S, Rowley CW, Colonius T, McKeon BJ, Schmidt OT, Gordeyev S, Theofilis V, Ukeiley LS. 2017 Modal analysis of fluid flows: An overview. *AIAA Journal* **55**, 4013–4041.
- [88] Schlueter-Kuck KL, Dabiri JO. 2017 Coherent structure colouring: Identification of coherent structures from sparse data using graph theory. *Journal of Fluid Mechanics* **811**, 468–486.